# SANFlow: Semantic-Aware Normalizing Flow for Anomaly Detection and Localization

**Daehyun Kim**[1]    **Sungyong Baik**[2]    **Tae Hyun Kim**[3]*

Dept. of Artificial Intelligence[1], Dept. of Data Science[2], Dept. of Computer Science[3]

Hanyang University

{daehyun, dsybaik, taehyunkim}@hanyang.ac.kr

## Abstract

Visual anomaly detection, the task of detecting abnormal characteristics in images, is challenging due to the rarity and unpredictability of anomalies. In order to reliably model the distribution of normality and detect anomalies, a few works have attempted to exploit the density estimation ability of normalizing flow (NF). However, previous NF-based methods forcibly transform the distribution of all features into a single distribution (e.g., unit normal distribution), even when the features can have locally distinct semantic information and thus follow different distributions. We claim that forcibly learning to transform such diverse distributions to a single distribution with a single network will cause the learning difficulty, thereby limiting the capacity of a network to discriminate between normal and abnormal data. As such, we propose to transform the distribution of features at each location of a given input image to different distributions. Specifically, we train NF to map the feature distributions of normal data to different distributions at each location in the given image. Furthermore, to enhance the discriminability, we also train NF to map the distribution of abnormal data to a distribution significantly different from that of normal data. The experimental results highlight the efficacy of the proposed framework in improving the density modeling and thus anomaly detection performance.

## 1   Introduction

Abnormal events that deviate significantly from expected and typical characteristics are often unwanted and are referred to as anomalies. The objective of anomaly detection is to identify such abnormal events. As such, anomaly detection is applicable across domains (e.g., video-surveillance, product defect detection, medical image analysis, etc.), where abnormal events can indicate or lead to critical issues. However, abnormal events occur rarely and can appear in various forms that cannot be known in advance, making it infeasible to collect a large amount of abnormal data in many real-world scenarios. The difficulty in collecting abnormal data has prevented anomaly detection from exploiting the recent breakthrough in supervised learning, leaving the field still challenging.

Acknowledging the difficulty of collecting abnormal data, many works have formulated anomaly detection as unsupervised learning, one-class learning, or few-shot learning [4, 5, 45, 49, 8, 47], in which the goal is to accurately model a distribution of normal data. Then, any data that deviates from the learned distribution of normal data is considered as an anomaly. The approaches to unsupervised anomaly detection differ by how they recognize the differences between normal and abnormal data. Reconstruction-based methods learn to reconstruct normal images and thus distinguish input images as abnormal if the reconstruction error is high [1, 42, 12, 35]. Meanwhile, representation-based approaches aim to learn a feature space, where normal images are brought close to each other

---

*Correspondence to: Tae Hyun Kim <taehyunkim@hanyang.ac.kr>.

37th Conference on Neural Information Processing Systems (NeurIPS 2023).

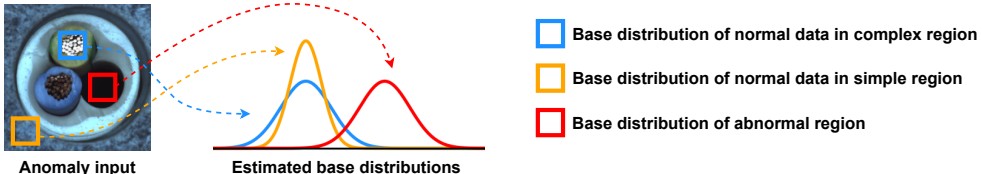

Figure 1: Motivation Overview: Different semantic features can be observed in different regions. Abnormal regions (red box) exhibit significantly distinct features compared to normal features (orange and blue boxes). Moreover, normal regions also display different semantic features based on the complexity of the region. Upon our observation, we propose Semantic-Aware Normalizing Flow (SANFlow) that enables more accurate density estimation by adaptively embedding semantic feature distribution into corresponding but different base distributions.

and abnormal images are placed far away from normal images [11, 8, 36]. Another line of works aims to simulate abnormalities through data augmentation [54, 26]. On the other hand, a recently emerging trend is to employ normalizing flow (NF) for a more reliable estimation of normal data distribution [51, 20, 37].

NF aims to achieve better density estimation by learning a sequence of reversible functions to map a complex distribution of input data into a simple distribution (e.g., normal distribution). The goal of NF aligns with that of anomaly detection in that a reliable density estimation of normal samples leads to the accurate detection of anomalies. However, previous NF-based methods have solely relied on the capability of NF to model complex distributions as simple ones. Although features at different locations, scales and images can follow different distributions, previous works have attempted to map such multi-modal distribution to simply a single normal distribution.

Considering that features at different locations and images may follow different distributions, we propose to adaptively embed the features at each location of each image into different distributions accordingly. We make two contributions in this work: First, we propose a novel NF-based framework that transforms the feature distribution of normal data at each location into Gaussian distributions with zero mean but different variances. The variance at each location is estimated by an external network that is conditioned on image features and semantics. Second, to enhance the detection of anomalies, we train NF to transform abnormal features into a distinct Gaussian distribution with a mean that is distant from the distributions of normal features. By embedding the features to locally different distributions, we facilitate the training of NF. We demonstrate that the estimated variance is lower for simple regions such as the background, while it increases for more complex regions. A description of the proposed approach is shown in Figure 1 and evaluation examples by category are shown in Figure 2.

The proposed framework demonstrates outstanding performance in both anomaly detection and localization. The strong empirical results underline the effectiveness of the proposed method in learning a more reliable density estimation, suggesting the importance of learning different target distributions for abnormal features and different normal features.

## 2 Related Work

Unsupervised anomaly detection tackles a challenging scenario, in which training images only consist of normal images that are free of anomalies. Under such challenging scenarios, standard supervised learning algorithms fail to work since training data is available for only one class (normal data), therefore often referred to as one-class learning. Methodologies that have emerged to overcome this issue largely differ by how they recognize the differences between normal data and abnormal data [9].

**Reconstruction based.** Reconstruction based approaches are hinged on the motivation that reconstruction models can reconstruct images well, if images are drawn from the same distribution as training images. Thus, these approaches distinguish images with high reconstruction error as abnormal. Several methods perform reconstruction by using auto-encoder architecture [41, 57, 6, 55, 32, 48, 45, 5] or the generator part of GAN models [1, 10, 31, 40].

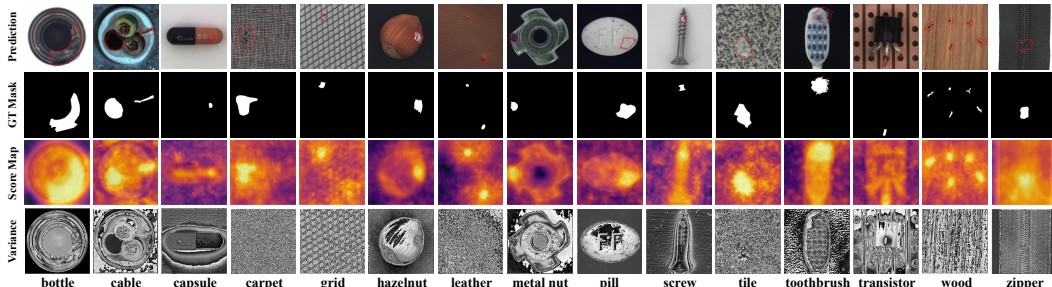

Figure 2: Anomaly localization results by our proposed framework on MVTec benchmark dataset. From top to bottom: anomaly localization prediction, GT mask, anomaly score map and estimated variance map of input image. Variance maps are derived from mode values of $\alpha$ and $\beta$ of each pixel and bright areas represent high variance and dark areas represent low variance values for corresponding distribution.

**Data augmentation based.** A few methods have proposed to generate abnormal images through data augmentation that is based on either cut-out [13], random erasing [43], noise generation [54], or geometric transformations [17, 16, 21]. In particular, CutPaste [26] introduces a new data augmentation to generate more fine-grained and subtle abnormal local patches in normal images. Furthermore, other research areas use synthesized abnormal images to improve performance such as semantic segmentation [23, 19, 7] or image classification [25].

**Representation based.** Another recent line of research has focused on designing systems that can leverage the strong representation power of pre-trained convolutional neural networks. [11, 36, 8, 49, 46, 56, 34]. Then, anomaly detection is performed by measuring the distances between test image features and normal image features. The methods differ by how the features of normal image are maintained or modeled and how the distance is measured. SPADE [8] employs k-nearest neighbor (k-NN) for normal image features retrieval and feature correspondence for measuring the differences. To mitigate the complexity issue with nearest neighbor search, PatchCore [36] maintains a subset of features via coreset subsampling. Meanwhile, PaDiM [11] models the normal features as a multivariate Gaussian distribution and uses Mahalanobis distance [30] as a distance measure, instead of the k-NN algorithm, to improve the inference complexity.

**Normalizing flow based.** Recently, a few works have shifted the attention to employing normalizing flow (NF) to better estimate the distribution of the features of normal patches. Through a sequence of learnable and invertible transformations, NF embeds the feature distribution into a simple normal distribution. DifferNet [37] is one of the first to employ NF to transform the distribution of CNN features to a normal distribution, where anomaly detection is performed at image level by measuring the likelihood of samples. Then, CFLOW-AD [20] extends NF-based methodology to pixel-level anomaly detection. FastFlow [51] further improves the performance by utilizing normalizing flow in two-dimensional space, unlike previous one-dimensional NF-based anomaly detection algorithms. In addition, CS-Flow [38] proposed a new kind of NF model that jointly processes multiple feature maps across different scales. However, they forcefully embed different feature distributions of semantic patches to a single normal distribution, despite recent findings in classification that it is beneficial to transform images of different semantics into different distributions [22, 3]. By contrast, we propose to train NF to transform the distributions at different locations to simple, but different distributions to better exploit the density estimation capability of NF by assuming locally varying base distributions.

## 3 Proposed method

In this section, we describe the overall pipeline of our framework, dubbed Semantic-Aware Normalizing Flow (SANFlow), as depicted in Figure 3. Given an image $x$, a pre-trained feature extractor $f$ is used to obtain features $v_i \sim V_i$ at the $i$-th position. Then, we employ normalizing flow (NF) $g^{-1} : V_i \to Z_i$ to embed the feature distribution $V_i$ (a.k.a. target distribution) to simple and well-known distribution $Z_i$ (a.k.a. base distribution). To further guide NF to map feature distributions $V_i$ into different base distributions $Z_i$ upon semantics, we introduce an external network $h$ which allows us to predict statistics of $z_i$ for each feature $v_i$. Moreover, similar to [26], we employ data

augmentation to synthesize local anomalies within the input image in order to train NF to embed the distribution of anomaly features into a distribution $Z_i^a$ that is distinct from the distributions of normal features $Z_i^n$.

Then, we modify the standard objective function of NF to account for different base distributions (Section 3.5). Finally, at test time, we use the trained NF to estimate the probability of given features to be normal, the negation of which is used as an anomaly score function to detect anomalies in a given image (Section 3.6).

## 3.1 Background on normalizing flow

Normalizing flow (NF) [33] aims to learn a function $g$ that transforms the latent variable $z$, which follows a simple and tractable base distribution $Z$, into the variable $v$, which follows a complex target distribution $V$. To do so, NF formulates $g$ as the composition of invertible functions $\{g^l\}_{l=1}^L$, i.e., $g = g^L \circ g^{L-1} \circ \cdots \circ g^1$, where $L$ is the number of invertible functions. Such transformation of a complex distribution into a simple one allows for efficient and exact density estimation, serving as a powerful tool in density estimation. By applying the change of variables theorem in a sequence, the log-likelihood of the target distribution $p_V(v)$ can be expressed in terms of the log-likelihood of the base distribution $p_Z(z)$:

$$\log p_V(v) = \log p_Z(z) - \sum_{l=1}^L \log \left| \det \frac{dg^l}{dz^{l-1}} \right|, \tag{1}$$

where $z^l$ represents the resulting random variable after applying up to the $l$-th function $g^l$ on $z$. Note that $z^0$ corresponds to $z$ while $z^L$ corresponds to $v$. For efficient computation of the log-likelihood, functions $\{g^l\}_{l=1}^L$ need to be invertible and have easy-to-compute Jacobian determinant. To meet such requirements, one of popular NF instances, RealNVP [14], formulated each function $g^l$ as an affine coupling block. In an affine coupling block, the input is first divided into two equal-dimension parts. Then, each part undergoes affine transformation, whose parameters are generated by a network conditioned on the other part. The transformed parts, in turn, are put back together via concatenation.

Then, the log-likelihood in Equation 1 is maximized (or the negative log-likelihood is minimized) to train the parameters of transformation function. A base distribution $Z$ is commonly chosen as a normal distribution $\mathcal{N}(0, I)$ in previous NF-based anomaly detection algorithms [20, 51, 37], resulting in a loss function (i.e., negative log-likelihood) as follows:

$$\mathcal{L}_{\text{NLL}} = \frac{||z||_2^2}{2} + \sum_{l=1}^L \log \left| \det \frac{dg^l}{dz^{l-1}} \right|. \tag{2}$$

## 3.2 Synthetic anomaly generation

In unsupervised anomaly detection scenarios, abnormal training images are typically not available. Therefore, we synthesize abnormal images to facilitate training of NF. Abnormal images usually differ from normal images only in local regions, which are semantically or structurally similar to surrounding normal regions. To generate such abnormal data, CutPaste [26] introduces data augmentation technique that disturbs local regions of normal images with semantically similar patches. The extracted patches, in turn, undergo transformations (e.g., random flip, rotate, blur, and color jittering) and replace the region of other normal images of the same category at random location. In this work, we use the CutPaste method to synthesize abnormal images. We modify this method slightly to generate more realistic abnormal images by blurring the borders of extracted patches and applying diverse color jittering values to these patches. Also, we make abnormal patches by taking small rectangular patches like CutPaste or newly used circular patches. Size of patches are random and from a normal image to retain diversity of abnormal regions and use the same ratio of all types of patches and normal data for training. Then, during training, our NF model handles normal and abnormal regions of each input image $x$ differently by assuming locally different base distributions. To achieve this, we introduce a binary mask $M$ as shown in Figure 3 for each synthetic anomaly image $x$ to denote whether each pixel at pixel location $(w, h)$ belongs to the normal region ($M_{w,h} = 1$) or the abnormal region ($M_{w,h} = 0$).

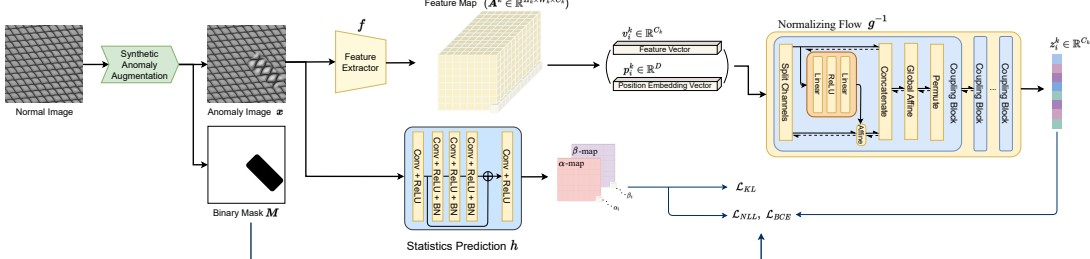

Figure 3: The overview of our proposed SANFlow framework. Firstly, with synthetic augmentation process, make binary mask $M$ and anomaly image $x$ from normal image. Then, a pre-trained feature extractor $f$ is applied to extract multi-scale feature maps ($K$=3, number of scales in this work) for $x$. For each $i$-th position at feature map of $k$-th level, each feature vector $v_i^k$ is concatenated with a position embedding vector $p_i^k$ before being fed into NF, which is independently trained for each scale. To be specific, NF consists of 8 coupling blocks (i.e., $L$=8), which maps given feature distributions to base distributions. Each base distribution is different in terms of statistics (variance in this work) which are dependent on semantic features of the corresponding location in the given image. In particular, we condition our statistics prediction network $h$ on the input image to generate parameters of inverse Gamma distribution such as $\alpha$ and $\beta$, to estimate the variances of base distributions, which are assumed to follow inverse Gamma distributions.

### 3.3 Feature extractor

Recently, features extracted by convolutional neural networks (CNN) pre-trained on ImageNet [39] have been shown to be helpful in anomaly detection, by providing useful semantic information [11, 5, 8, 20, 48]. In particular, several methods have employed a multi-scale feature pyramid [27] to handle anomalies of diverse sizes [20, 48, 8]. This is because features of different scales capture information about regions of corresponding sizes (i.e., effective receptive field) in the image [29]. Motivated by previous findings, we also employ a pre-trained CNN [11, 20, 36] to obtain a $K$-level feature pyramid. The feature map at the $k$-th scale is denoted by $A^k \in \mathbb{R}^{C_k \times H_k \times W_k}$ where $C_k$ is the number of channels and $H_k$ and $W_k$ are the height and width of the feature maps, respectively.

### 3.4 Semantic-aware normalizing flow

#### 3.4.1 Scale- and spatial-aware normalizing flow

We train NF to map our feature vectors computed by CNN to latents which follow simple base distributions. However, features can follow complex distributions at each scale and spatial location of feature map. Thus, using a single NF model to map such complex features to a single base distribution can be difficult. Aiming to bridge the gap, CFLOW-AD [20] applied two modifications to a standard NF baseline model RealNVP [14]: independent NF models for each scale and conditioning NF on position embedding vector [15, 44], similar to [2]. Inspired by CFLOW-AD, we also employ $K$ independent NF models to handle features across different scales as we have $K$-level pyramid features and position embedding vector conditioning. Moreover, we train the NF model to map features to latents that follow a spatially varying underlying distribution. This allows NF to consider semantic information, such as feature complexity, for corresponding locations in the image.

To be specific, we create a set of vectors by taking a feature vector $v_i^k$ at the $i$-th position (i.e., 2D coordinate) of the $k$-th scale feature map $A^k$, as done in the previous work [20]. Note that, we omit the scale index $k$ from this point on to avoid cluttered notation. To retain the spatial information, each feature vector $v_i$ is concatenated with the corresponding position embedding vector $p_i \in \mathbb{R}^D$, whose values are composed of sinusoidal harmonics and unique to each $i$-th position [15, 44]. The resulting vector $c_i = (v_i, p_i)$ is now used to condition each transformation function $g^l$ in NF, making it process position-aware [2]. Then, the log-likelihood of feature distribution becomes:

$$\log p_{V_i}(v_i) = \log p_{Z_i}(z_i) - \sum_{l=1}^{L} \log \left| \det \frac{dg^l}{dz_i^{l-1}} \right|, \qquad (3)$$

where $Z_i$ is a simple, yet locally varying base distribution corresponding to a feature vector $\boldsymbol{v}_i$.

However, just making NF scale- and spatial-aware is insufficient to handle variations that exist within the same object category. This is because objects are not perfectly aligned across different images of the same category, resulting in images with different semantic information even at the identical spatial location and scale. Thus, in order to enhance the density estimation of our NF model, we propose to embed feature distributions into different base distributions, leveraging the semantic information encoded in the features. Further details on this approach are provided in the following section.

### 3.4.2 Semantic-aware base distribution

In this study, we instantiate semantic-dependent base distributions as Gaussian distributions with statistics that are estimated based on semantic information of features. Inspired by an algorithm that estimates the statistics of non-i.i.d. noise [52], we employ a lightweight statistics prediction network $h$ to estimate the statistics for the given feature $\boldsymbol{v}_i$. We condition the network, parameterized by $\boldsymbol{\theta}_h$, on the semantic features to be able to estimate appropriate statistics of corresponding base distributions. As we will discuss in our experiments, we empirically observe that estimating both mean ($\mu_i$) and variance ($\sigma_i^2$) of the base distribution is difficult and that estimating variance ($\sigma_i^2$) alone is beneficial (Section 4.3). Therefore, we fix $\mu_i$ to be 0 for normal regions and 1 for abnormal regions. In doing so, we map abnormal features to a base distribution with minimal overlap with that of normal features during training, thereby aiding NF in mapping normal and abnormal regions into distinct distributions. When samples are non-i.i.d., the distribution of sample variance can be assumed to follow the inverse Gamma distribution [52]. Since image pixels and semantic features are non-i.i.d., we formulate the variances of the corresponding base distributions to follow the inverse Gamma distribution as $\text{IG}(\cdot|\alpha, \beta)$ with parameters $\alpha$ and $\beta$. We set $\alpha$ to be $(\frac{p^2}{2} - 1)$ and $\beta$ to be $\frac{p^2 \xi}{2}$, where $p^2$ denotes the area of the corresponding receptive field of size $p \times p$ and $\xi$ is the mode of the inverse Gamma distribution [52, 50]. In this work, $\xi$ is a hyperparameter, the value of which is empirically found to be 0.1. Upon the assumption, we train the network $h$ to estimate $\alpha_i$ and $\beta_i$ for each $\boldsymbol{v}_i$, resulting in the following regularization loss as:

$$
\begin{aligned}
\mathcal{L}_{KL} &= D_{KL}(IG(\alpha_i, \beta_i) \,|\, IG(\alpha, \beta)) \\
&= \sum_{i=1}^{H_k \times W_k} \{(\alpha_i - \alpha)\psi(\alpha_i) + (\log \Gamma(\alpha) - \log \Gamma(\alpha_i)) + \alpha(\log \beta_i - \log \beta) + \alpha_i(\frac{\beta}{\beta_i} - 1)\},
\end{aligned}
\tag{4}
$$

where $D_{KL}(IG(\alpha_i, \beta_i) \,|\, IG(\alpha, \beta))$ computes a KL divergence between the estimated distribution by the network $h$ and the distribution of variance $\sigma_i^2$, which we assume to follow $IG(\alpha, \beta)$ as mentioned above; $\psi$ is a digamma function and $\Gamma$ is a gamma function. Notably, $\mathcal{L}_{KL}$ with $\alpha$ and $\beta$ acts as regularization to guide the statistics prediction network for stable training. The derivation details and detailed explanations can be found in the supplementary material.

### 3.5 Loss function

In this section, we delineate the overall objective function which our model is trained to minimize. First, we need to modify the log-likelihood of $\boldsymbol{z}_i$ corresponding to feature $\boldsymbol{v}_i$ in Equation 3 to account for handling both normal and abnormal features differently:

$$
\log p_{Z_i}(\boldsymbol{z}_i) = m_i \cdot \log p_{Z_i^n}(\boldsymbol{z}_i) + (1 - m_i) \cdot \log p_{Z_i^a}(\boldsymbol{z}_i),
\tag{5}
$$

where the binary indicator $m_i$ is set to be 1 when the corresponding location in the binary mask $\boldsymbol{M}$ is 1, and 0, otherwise. Note that, $Z_i^n$ represents the corresponding base distribution when $\boldsymbol{v}_i$ is a normal feature, while $Z_i^a$ represents the base distribution when $\boldsymbol{v}_i$ is an abnormal feature. In practice, to compute the likelihood at the $k$-th scale, the binary mask $\boldsymbol{M}$ is resized to match the size of the feature map $\boldsymbol{A}^k$, which has height $H_k$ and width $W_k$. The resizing is done using nearest-neighbor interpolation to preserve the binary nature of the mask. In turn, each log-likelihood term can be formulated as follows (detailed derivations can be found in the supplementary material):

$$\log p_{Z_i^n}(\boldsymbol{z}_i) = -\frac{1}{2}\log 2\pi - \frac{1}{2}(\log \beta_i - \psi(\alpha_i)) - \frac{\alpha_i}{2\beta_i}||\boldsymbol{z}_i||_2^2, \tag{6}$$

and,

$$\log p_{Z_i^a}(\boldsymbol{z}_i) = -\frac{1}{2}\log 2\pi - \frac{1}{2}(\log \beta_i - \psi(\alpha_i)) - \frac{\alpha_i}{2\beta_i}||\boldsymbol{z}_i - 1||_2^2. \tag{7}$$

Moreover, to further aid NF in distinguishing normal and abnormal features, we can also train NF to perform binary classification using the loss function as follows:

$$\mathcal{L}_{\mathrm{BCE}} = \mathbb{BCE}(s(\boldsymbol{z}_i), m_i), \tag{8}$$

where $\mathbb{BCE}$ denotes the conventional binary cross-entropy loss, while $s(\boldsymbol{z}_i)$ measures the probability of $\boldsymbol{z}_i$ being classified as normal (i.e., $s(\boldsymbol{z}_i) = \frac{p_{Z_i^n}(\boldsymbol{z}_i)}{p_{Z_i^n}(\boldsymbol{z}_i) + p_{Z_i^a}(\boldsymbol{z}_i)}$). Note that the log-likelihood of $\boldsymbol{z}_i$ being abnormal (Equation 7) can be used only during training such that our framework learns to map abnormal features to base distribution $Z^a$ that has small overlap with the base distributions of normal features $Z^n$. This allows NF to transform normal and abnormal features to two distinct base distributions, thereby enhancing the discriminability.

Together with $\mathcal{L}_{\mathrm{KL}}$ from Equation 4, the overall objective function $\mathcal{L}_{\mathrm{overall}}$ to minimize is given by,

$$\mathcal{L}_{\mathrm{overall}} = \mathcal{L}_{\mathrm{NLL}} + \lambda_1 \cdot \mathcal{L}_{\mathrm{BCE}} + \lambda_2 \cdot \mathcal{L}_{\mathrm{KL}}, \tag{9}$$

where $\lambda_1$ and $\lambda_2$ are hyperparameters to adjust the associated weights of $\mathcal{L}_{\mathrm{BCE}}$ and $\mathcal{L}_{\mathrm{KL}}$.

### 3.6 Anomaly score functions

During inference, given a test image $\boldsymbol{x} \in \mathbb{R}^{H \times W \times 3}$, we first compute the log-likelihood of each feature $\boldsymbol{v}_i$ at each $k$-th scale via Equation 3, where the log-likelihood of $\boldsymbol{z}_i$ is computed with Equation 6. Then, the likelihood map $\boldsymbol{P}^k \in \mathbb{R}^{H_k \times W_k}$ for the $k$-th scale is obtained by taking exponential of the log-likelihood 6. We obtain the final likelihood map $\boldsymbol{P} \in \mathbb{R}^{H \times W}$ by summing the likelihood map $\boldsymbol{P}^k$ from all scales. These likelihood maps are first upsampled to the image resolution $\mathbb{R}^{H \times W}$ using bilinear interpolation, following the approach described in [20]. Finally, the anomaly score map is computed as the negative of $\boldsymbol{P}$, resulting in $-\boldsymbol{P}$.

## 4 Experiments

In this section, we evaluate our framework to validate its capability in both pixel-level anomaly localization and image-level anomaly detection. We will release our code and data upon acceptance, and more details and results can be found in the supplementary material.

### 4.1 Experimental settings

**Dataset.** The experiments are conducted on two commonly used datasets for unsupervised anomaly detection: STC (ShanghaiTech Campus) dataset [28] and MVTec dataset [4]. STC is a video surveillance dataset, which provides static videos of 13 different scenes of $856 \times 480$ resolution. It contains $274,515$ frames for training and $42,883$ frames for evaluation. The training set consists of only normal sequences, while the evaluation set consists of $300,308$ regular frames and $42,883$ irregular frames. MVTec is a dataset that consists of images of industrial products categorized into 5 texture categories and 10 object categories. To evaluate the unsupervised anomaly detection performance, the training set includes only defect-free (e.g., normal) images: $3,629$ normal images are available for training while $1,725$ normal and abnormal images are used as test set. Among the test images, $1,258$ images contain defects (e.g., abnormal images). For data augmentation, abnormal patches undergo random vertical, horizontal flip and rotation during training, as described in 3.2. We evaluate and compare algorithms in terms of area under the receiver operating characteristic curve (AUROC) and area under the per-region-overlap curve (AUPRO), as used in [4, 20]. AUPRO scores can be found in the supplementary.

Table 1: Anomaly detection and localization results w.r.t. AUROC metric by various methods with WRN-50 backbone for feature extraction on STC [28].

| | FramePred [28] | MemAE [18] | SPADE [8] | PaDiM [11] | CFLOW-AD [20] | PatchCore-10 [36] | SANFlow (Ours) |
|---|---|---|---|---|---|---|---|
| Image-wise | 72.8 | 71,2 | 71.9 | - | 72.63 | - | **76.1** |
| Pixel-wise | - | - | 89.9 | 91.2 | 94.48 | 91.8 | **94.8** |

Table 2: Quantitative comparisons on anomaly localization performace in MVTec dataset [4] with respect to AUROC metric. For each category, the best and second best performance is **bolded** and underlined, respectively.

| Category | $AE_{SSIM}$ WRN-50 | $\gamma$-VAE+grad WRN-50 | PatchSVDD WRN-50 | PaDiM WRN-50 | CutPaste WRN-50 | CFLOW-AD WRN-50 | PatchCore-10 WRN-50 | PatchCore-1 WRN-101 | **SANFlow** WRN-50 | **SANFlow** WRN-101 |
|---|---|---|---|---|---|---|---|---|---|---|
| bottle | 93.0 | 93.1 | 98.1 | 98.3 | 97.6 | 98.76 | 98.6 | 98.6 | 98.6 | **99.1** |
| cable | 82.0 | 88.0 | 96.8 | 96.7 | 90.0 | 97.64 | 98.5 | 98.4 | 98.5 | **98.8** |
| capsule | 94.0 | 91.7 | 95.8 | 98.5 | 97.4 | 98.98 | 98.9 | **99.1** | **99.1** | 98.9 |
| carpet | 87.0 | 72.7 | 92.6 | 99.1 | 98.3 | 99.23 | 99.1 | 98.7 | 99.3 | **99.4** |
| grid | 94.0 | 97.9 | 96.2 | 97.3 | 97.5 | 96.89 | 98.7 | 98.7 | 98.5 | **99.3** |
| hazelnut | 97.0 | 98.8 | 97.5 | 98.2 | 97.3 | 98.82 | 98.7 | 98.8 | **99.2** | 99.0 |
| leather | 78.0 | 89.7 | 97.4 | 99.2 | 99.5 | 99.61 | 99.3 | 99.3 | 99.6 | **99.8** |
| metal nut | 89.0 | 91.4 | 98.0 | 97.2 | 93.1 | 98.56 | 98.4 | **98.8** | 98.5 | 98.7 |
| pill | 91.0 | 93.5 | 95.1 | 95.7 | 95.7 | 98.95 | 97.6 | 97.8 | **99.2** | 99.1 |
| screw | 96.0 | 97.2 | 95.7 | 98.5 | 96.7 | 98.10 | **99.4** | 99.3 | 99.0 | 99.2 |
| tile | 59.0 | 58.1 | 91.4 | 94.1 | 90.5 | 97.71 | 95.9 | 96.1 | 98.9 | **99.1** |
| toothbrush | 92.0 | 98.3 | 98.1 | 98.8 | 98.1 | 98.56 | 98.7 | 98.8 | 98.9 | **99.2** |
| transistor | 90.0 | 93.1 | 97.0 | **97.5** | 93.0 | 93.28 | 96.4 | 96.4 | 94.4 | 95.1 |
| wood | 73.0 | 80.9 | 90.8 | 94.9 | 95.5 | 94.49 | 95.1 | 95.1 | 96.4 | **97.9** |
| zipper | 88.0 | 87.1 | 95.1 | 98.5 | 99.3 | 98.41 | 98.9 | 98.9 | 98.9 | **99.6** |
| average | 87.0 | 88.8 | 95.7 | 97.5 | 96.0 | 97.9 | 98.1 | 98.2 | 98.5 | **98.8** |

Table 3: Quantitative comparisons on anomaly detection in MVTec dataset [4] with respect to AUROC metric. For each category, the best and second best performance is **bolded** and underlined, respectively.

| Category | GANomaly WRN-50 | OCSVM WRN-50 | PatchSVDD WRN-50 | DifferNet WRN-50 | PaDiM WRN-50 | CFLOW-AD WRN-50 | CutPaste WRN-50 | PatchCore-10 WRN-50 | PatchCore-1 WRN-101 | **SANFlow** WRN-50 | **SANFlow** WRN-101 |
|---|---|---|---|---|---|---|---|---|---|---|---|
| bottle | 89.2 | 99 | 98.6 | 99.0 | - | **100** | 98.2 | **100** | **100** | **100** | **100** |
| cable | 75.7 | 80.3 | 90.3 | 95.9 | - | 97.59 | 81.2 | 99.4 | 99.6 | 99.4 | **99.7** |
| capsule | 73.2 | 54.4 | 76.7 | 86.9 | - | 97.68 | 98.2 | 97.8 | 98.2 | 97.7 | **98.9** |
| carpet | 69.9 | 62.7 | 92.9 | 92.9 | - | 98.73 | 93.9 | 98.7 | 98.4 | 99.8 | **99.9** |
| grid | 70.8 | 41 | 94.6 | 84.0 | - | 99.60 | **100** | 97.9 | 99.8 | 99.3 | **100** |
| hazelnut | 78.5 | 91.1 | 92.0 | 99.3 | - | 99.98 | 98.3 | **100** | **100** | **100** | **100** |
| leather | 84.2 | 88 | 90.9 | 97.1 | - | **100** | **100** | **100** | **100** | **100** | **100** |
| metal nut | 70.0 | 61.1 | 94.0 | 96.1 | - | 99.26 | 99.9 | **100** | **100** | 99.8 | **100** |
| pill | 74.3 | 72.9 | 86.1 | 88.8 | - | 96.82 | 94.9 | 96.0 | 97.2 | 96.8 | **98.2** |
| screw | 74.6 | 74.7 | 81.3 | 96.3 | - | 91.89 | 88.7 | **98.9** | 97.0 | 94.0 | 96.2 |
| tile | 79.4 | 87.6 | 97.8 | 99.4 | - | 99.88 | 94.6 | 98.9 | 98.9 | **100** | **100** |
| toothbrush | 65.3 | 61.9 | **100** | 98.6 | - | 99.65 | 99.4 | 99.7 | **100** | 96.7 | **100** |
| transistor | 79.2 | 56.7 | 91.5 | 91.1 | - | 95.21 | 96.1 | **100** | **100** | 99.3 | 99.4 |
| wood | 83.4 | 95.3 | 96.5 | **99.8** | - | 99.12 | 99.1 | 99.0 | 99.5 | 99.1 | 99.3 |
| zipper | 74.5 | 51.7 | 97.9 | 95.1 | - | 98.48 | 99.9 | 99.5 | 99.9 | 99.9 | **100** |
| average | 76.2 | 71.9 | 92.1 | 94.9 | 97.9 | 98.26 | 96.1 | 99.0 | **99.4** | 98.7 | **99.4** |

**Implementation details.** Following other works [4, 20, 36], our models are trained and evaluated separately for each category. Similar to CFLOW-AD [20], we use $K$=3 scales for feature pyramid; normalizing flow consists of $L$=8 transformation blocks; the dimension of position embedding vector $D$ is 512; Adam optimizer with a learning rate of 5e-4 and 80 train epochs for training. The loss weight hyperparameters $\lambda_1$ and $\lambda_2$ are set to be 1.0 and 0.2, respectively. The statistics prediction network $h$ consists of five convolutional layers with a skip connection as illustrated in Figure 3. At last, we use NVIDIA RTX8000 for training and the model shows near real-time performance by running at 13fps.

## 4.2 Experimental results

We provide the anomaly localization and detection performance on the STC dataset in Table 1 while the results on MVTec dataset are reported in Table 2 and Table 3. All results presented in this study are obtained using the same backbone architecture, WRN-50 [53], while additional results on MVTec are reported for PatchCore [36] and our method with a larger backbone, WRN-101 [53]. The experimental results demonstrate that the performance of our proposed method is comparable with other state-of-the-art methods. In Table 1, compare to other anomaly detection models that are not video targeted, our method shows outstanding performances in both detection and localization.

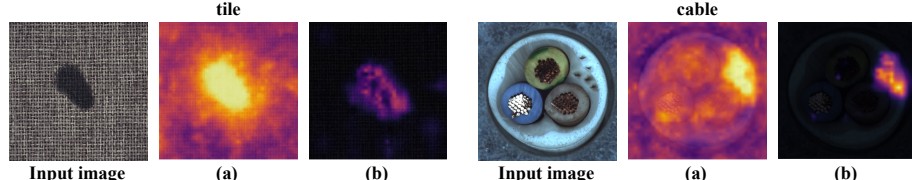



| tile | | | cable | | |
|------|------|------|------|------|------|
| Input image | (a) | (b) | Input image | (a) | (b) |



Figure 4: Visualization of anomaly score map based on **(a)** $Z^n$ and **(b)** $Z^a$ on MVTec dataset.

Table 4: Ablation study on loss functions in MVTec and STC datasets.

|  | $\mathcal{L}_{\text{NLL}}$ | $\mathcal{L}_{\text{BCE}}$ | $\mathcal{L}_{\text{KL}}$ | STC | MVTec |
|---|---|---|---|---|---|
| Model **(1a)** | Eq.(2) | ✗ | ✗ | 72.6 | 98.3 |
| Model **(1b)** | Eq.(3)&(5) | ✗ | ✗ | 73.5 | 98.2 |
| Model **(1c)** | Eq.(3)&(5) | ✓ | ✗ | 73.1 | 98.3 |
| Model **(1d)** | Eq.(3)&(5) | ✗ | ✓ | 74.2 | 98.5 |
| **SANFlow** | Eq.(3)&(5) | ✓ | ✓ | **76.1** | **98.7** |

Table 5: Ablation study on anomaly augmentation in MVTec dataset.

|  | Augmentation | $\mathcal{L}_{\text{NLL}}$ | $\mathcal{L}_{\text{BCE}}$ | $\mathcal{L}_{\text{KL}}$ | STC | MVTec |
|---|---|---|---|---|---|---|
| Model **(1d)** | ✓ | ✓ | ✗ | ✓ | 74.2 | 98.5 |
| Model **(2a)** | ✗ | ✓ | ✗ | ✓ | 73.6 | 98.4 |
| **SANFlow** | ✓ | ✓ | ✓ | ✓ | **76.1** | **98.7** |

Plus, Table 2 and Table 3 reports that our proposed method performs better than existing NF-based anomaly detection algorithms (CFLOW-AD [20] and DifferNet [37]) and other anomaly detection algorithms such as $AE_{SSIM}$ [4], $\gamma$-VAE+grad [12], PatchSVDD [49], PaDiM [11] and PatchCore-1, 10 [36]. In particular, the results demonstrate the effectiveness of our proposed semantic-aware base distribution mapping approach in enhancing the density estimation performance of the proposed NF model.

## 4.3 Ablation study

We conduct ablation experiments to evaluate the efficacy of each component of our framework. Specifically, we perform ablation studies on the loss function (Table 4), synthetic anomaly generation (Table 5), and statistics estimated by our statistics prediction network $h$ (Table 6). All ablation experiments are reported with respect to AUROC, using the WRN-50 backbone, and all average AUROC values over all categories are reported for image-wise anomaly detection.

In Table 4, we analyze the efficacy of our proposed loss functions. To do so, we first compare against a baseline NF model trained using the standard negative log-likelihood loss of NF in Equation 2 (Model **(1a)**). Compared to Model **(1a)**, SANFlow brings substantial improvement, thereby validating the effectiveness of all three proposed loss functions ($\mathcal{L}_{\text{NLL}}$ in Equation 3 using Equation 5, $\mathcal{L}_{\text{KL}}$, and $\mathcal{L}_{\text{BCE}}$). We validate the efficacy of each loss function separately by applying only $\mathcal{L}_{\text{NLL}}$ using Equation 3 and 5 (Model **(1b)**) or disabling either $\mathcal{L}_{\text{KL}}$ (Model **(1c)**) or $\mathcal{L}_{\text{BCE}}$ (Model **(1d)**). While each variant leads to performance improvement, using all three proposed loss functions leads to the largest improvement, suggesting that all three proposed loss functions coordinate and play critical roles in our overall framework.

In Table 5, we evaluate the impact of synthetic anomaly augmentation (generation) on the final performance of our framework. Training without anomaly augmentation and thus without any anomaly data results in performance degradation (Model **(2a)**). Also, Model **(1d)** (Table 4) which has only difference at doing anomaly augmentation in training, performs better than Model **(2a)**. Thus, the ablation result suggests that not only embedding semantic features to different base distributions but training NF with synthetic anomalies is also an important process. For better understanding, Fig-

Table 6: Ablation study on statistics estimation in MVTec and STC datasets. **fixed/estimated** refers to whether statistics are manually fixed or estimated by our statistics prediction network.

|  | $\mu_i$ | $\sigma_i^2$ | STC | MVTec |
|---|---|---|---|---|
| Model **(3a)** | fixed | fixed | 75.9 | 97.6 |
| Model **(3b)** | estimated | fixed | 75.1 | 97.0 |
| Model **(3c)** | estimated | estimated | 74.5 | 98.1 |
| **SANFlow (Ours)** | fixed | estimated | **76.1** | **98.7** |

ure 4 visualizes anomaly score maps based on estimated normal distribution $Z^n$ **(a)** and abnormal distribution $Z^a$ **(b)** for test images. As described in Section 3.6, Figure 4 **(a)** can be found with Equation 3 and 6 while Figure 4 **(b)** can be found with Equation 3 and 7. We note that both maps agree on the abnormal region (highlighted with bright colors), validating that training NF to map normal and abnormal features to distant base distributions helps the model discriminate abnormal regions from normal regions.

Table 6 reports ablation results after fixing either mean ($\mu_i$), variance ($\sigma_i^2$), or both statistics of base distributions. If $\mu_i$ is fixed, it is set to be 0 for normal features and 1 for abnormal features. If $\sigma_i^2$ is fixed, it is empirically set to be 0.1 for both normal and abnormal features. The performance is observed to decrease noticeably when $\mu_i$ is estimated alone (Model **(3b)**) or together with $\sigma_i^2$ (Model **(3c)**). On the other hand, estimating only $\sigma_i^2$ while fixing $\mu_i$ brings significant performance improvement. The performance degradation from estimating $\mu_i$ may be due to learning complexity. Furthermore, to enhance the persuasiveness of the mean margin, we conducted additional experiments on the MVTec dataset, comparing performance at margin values of 0.5, 1.0, and 1.5 in WRN-50 backbone. For margin values 0.5, 1.0, and 1.5, the proposing method shows 98.6, 98.7, and 98.1 AUROC for image-wise and 98.4, 98.5, and 98.3 AUROC for pixel-wise. As the margin of 1.0 yields the highest results in both image- and pixel- wise, we could confirm that the mean margin of 1.0 is a persuasive choice.

## 4.4   Experiments on other datasets

Although the MVTec dataset is a benchmark that is widely used for anomaly detection and localization, there is difficulty in identifying the advantage of the proposing method with it due to performance saturation. Therefore, we provide a performance comparison in the VisA dataset [58]. Our proposed method with a WRN-50 backbone achieves an image-wise AUROC of 93.4, 98.6, and a PRO score of 89.4, while CFLOW-AD [20] and PaDiM [11] achieve 91.5, 59.8, and 89.1, 85.9 of an image-wise AUROC and a PRO score, respectively. The results show that our approach demonstrates competitive performance against other methods, including other flow-based approaches, demonstrating the effectiveness of our framework in greatly improving the density estimation capability of normalizing flow for anomaly detection and localization. Plus, while our approach excels at detecting anomalies within images, its effectiveness in tasks involving image-wise semantic outlier detection, like CutPaste [26] and CFLOW-AD, is limited. Nonetheless, we conducted an additional experiment with CIFAR-10 [24], where our proposed method achieved an AUROC of 80.8, outperforming performances of CutPaste (69.4) and CFLOW-AD (79.32).

## 5   Conclusion

In this work, we propose to improve the density estimation of normalizing flow (NF) for anomaly detection, by training NF to dynamically embed given feature distribution to different base distributions. In particular, base distributions are Gaussian distribution with statistics estimated by our statistics prediction network. As a result, our NF framework learns to map not only diverse normal features but also abnormal features to corresponding yet different base distributions, enhancing the density estimation capability.

**Limitation** While SANFlow has the capability of mapping different semantic features to different base distributions, it requires anomaly augmentation to map anomaly features to a base distribution distant from base distributions of normal features for better anomaly detection. However, augmentations require domain knowledge and do not cover new anomalies that may occur. But, we would like to note that our ablation study reveals that our framework brings improvements even without anomaly augmentation. Nonetheless, reducing the dependence on anomaly augmentations in the proposed framework is an interesting future research direction.

**Broader Impacts** Finding anomalies in product during fabrication is important to ensure the quality of products. Anomaly detection algorithms can automate the process, which can offload the burden from human workers. This will allow human workers to focus on other important aspects of fabrication, improving the overall process of fabrication and thus the quality of products. Furthermore, anomaly detection can be used to frequently check anomalies of objects that may be difficult for humans but crucial to human safety (e.g., tall buildings, bridges, nuclear power plants, etc.).

**Acknowledgements** This work was supported by Institute of Information & communications Technology Planning & Evaluation (IITP) grant funded by the Korea government(MSIT) (No. RS-2023-00220628, Artificial intelligence for prediction of structure-based protein interaction reflecting physicochemical principles, 50%) and Samsung Electronics Co., Ltd, and Samsung Research Funding Center of Samsung Electronics (No. SRFCIT1901- 06, 50%).

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

# A  Loss Function Derivation

In this section, we show the detailed derivations for Equation 4 in Section 3.4.2 and Equation 6 and 7 in Section 3.5.

## A.1  KL-Divergence

We present the detailed derivations of our KL divergence loss $\mathcal{L}_{\text{KL}}$, where $IG(\alpha_i, \beta_i)$ and $IG(\alpha, \beta)$ represent the probability density function of the inverse Gamma distribution with their corresponding parameters $\alpha_i, \beta_i$ and $\alpha, \beta$. Furthermore, $\boldsymbol{s}_i$ and $\boldsymbol{s}$ denote random variables that follow the corresponding distributions $IG(\alpha_i, \beta_i)$ and $IG(\alpha, \beta)$, respectively. The detailed formulations are as follows:

$$
\begin{aligned}
\mathcal{L}_{KL} &= D_{KL}(IG(\alpha_i, \beta_i) \,|\, IG(\alpha, \beta)) \\
&= \sum_{i=1}^{H_k \times W_k} E[\log IG(\alpha_i, \beta_i) - \log IG(\alpha, \beta)] \\
&= \sum_{i=1}^{H_k \times W_k} \{E[-\log IG(\alpha, \beta)] - E[-\log IG(\alpha_i, \beta_i)]\} \\
&= \sum_{i=1}^{H_k \times W_k} \{E[-\alpha \log \beta + \log \Gamma(\alpha) + (\alpha+1)\log \boldsymbol{s} + \frac{\beta}{\boldsymbol{s}}] \\
&\qquad\qquad - E[-\alpha_i \log \beta_i + \log \Gamma(\alpha_i) + (\alpha_i+1)\log \boldsymbol{s}_i + \frac{\beta_i}{\boldsymbol{s}_i}]\} \\
&= \sum_{i=1}^{H_k \times W_k} \{[\alpha + \log \beta \Gamma(\alpha) - (\alpha+1)\psi(\alpha)] - [\alpha_i + \log \beta_i \Gamma(\alpha_i) - (\alpha_i+1)\psi(\alpha_i)]\} \\
&= \sum_{i=1}^{H_k \times W_k} \{(\alpha_i - \alpha)\psi(\alpha_i) + (\log \Gamma(\alpha) - \log \Gamma(\alpha_i)) + \alpha(\log \beta_i - \log \beta) + \alpha_i(\frac{\beta}{\beta_i} - 1)\}.
\end{aligned}
$$

$$(\text{S1})$$

## A.2  Log-likelihood Derivation

To calculate the semantic-aware variance $\sigma_i$ for log-likelihood computation, we approximate it using the mode value ($\sigma_i$) from the inverse Gamma distribution $IG(\alpha_i, \beta_i)$. This approximation is employed for computational efficiency, following the approach used in VDNet [52]. Therefore, in Equation S2 and Equation S3, we use $h(\sigma_i^2|\boldsymbol{x})$ that predicts statistics of semantic-aware variances (i.e., $\sigma_i$) from input image $\boldsymbol{x}$ as:

$$
\begin{aligned}
\log p_{Z_i^n}(\boldsymbol{z}_i) &= E_{h(\sigma_i^2|\boldsymbol{x})}[\log p(\boldsymbol{z}_i|0, \sigma_i^2)] \\
&= \int h(\sigma_i^2|\boldsymbol{x}) \log p(\boldsymbol{z}_i|0, \sigma_i^2) d\sigma^2 \\
&= \int h(\sigma_i^2|\boldsymbol{x})\{-\frac{1}{2}\log 2\pi - \frac{1}{2}\log \sigma_i^2 - \frac{||\boldsymbol{z}_i||_2^2}{2\sigma_i^2}\} d\sigma^2 \\
&= -\frac{1}{2}\log 2\pi - \frac{1}{2}\int h(\sigma_i^2|\boldsymbol{x})\log \sigma_i^2 d\sigma_i^2 - \frac{1}{2}||\boldsymbol{z}_i||_2^2 \int h(\sigma_i^2|\boldsymbol{x})\frac{1}{\sigma_i^2} d\sigma_i^2 \\
&= -\frac{1}{2}\log 2\pi - \frac{1}{2}E[\log \sigma_i^2] - \frac{1}{2}||\boldsymbol{z}_i||_2^2 E[\frac{1}{\sigma_i^2}] \\
&= -\frac{1}{2}\log 2\pi - \frac{1}{2}(\log \beta_i - \psi(\alpha_i)) - \frac{\alpha_i}{2\beta_i}||\boldsymbol{z}_i||_2^2,
\end{aligned}
$$

$$(\text{S2})$$

and

$$\log p_{Z_i^a}(\boldsymbol{z}_i) = E_{h(\sigma_i^2|\boldsymbol{x})}[\log p(\boldsymbol{z}_i|1, \sigma_i^2)]$$

$$= \int h(\sigma_i^2|\boldsymbol{x}) \log p(\boldsymbol{z}_i|1, \sigma_i^2) d\sigma^2$$

$$= \int h(\sigma_i^2|\boldsymbol{x})\{-\frac{1}{2}\log 2\pi - \frac{1}{2}\log \sigma_i^2 - \frac{||\boldsymbol{z}_i - 1||_2^2}{2\sigma_i^2}\}d\sigma^2$$

$$= -\frac{1}{2}\log 2\pi - \frac{1}{2}\int h(\sigma_i^2|\boldsymbol{x}) \log \sigma_i^2 d\sigma_i^2 - \frac{1}{2}||\boldsymbol{z}_i - 1||_2^2 \int h(\sigma_i^2|\boldsymbol{x})\frac{1}{\sigma_i^2}d\sigma_i^2 \quad \text{(S3)}$$

$$= -\frac{1}{2}\log 2\pi - \frac{1}{2}E[\log \sigma_i^2] - \frac{1}{2}||\boldsymbol{z}_i - 1||_2^2 E[\frac{1}{\sigma_i^2}]$$

$$= -\frac{1}{2}\log 2\pi - \frac{1}{2}(\log \beta_i - \psi(\alpha_i)) - \frac{\alpha_i}{2\beta_i}||\boldsymbol{z}_i - 1||_2^2.$$

## B   Implementation Details

### B.1   Hyperparameters

As described in the main manuscript, our overall framework comprises a pre-trained feature extractor $f$, a normalizing flow $g$, and a statistics prediction network $h$. For the feature extraction process, we utilize a multi-scale feature pyramid approach. Specifically, we extract features from three different scales ($K = 3$) using a pre-trained feature extractor. Regarding the normalizing flow model, we employ $L = 8$ transformation blocks for each scale.

Similar to VDNet [52], our statistics prediction network takes an image as input and consists of five convolutional layers and a skip-connection. Each convolutional layer comprises 64 filters wit a size of $3 \times 3$ except for the last layer. In particular, the last layer of the statistics prediction network produces a two-channel output: one for $\alpha$ and the other for $\beta$.

During training phase on the MVTec dataset [4], we fixed the learning rate to 5e-4. Moreover, we empirically determined the hyperparameters $\lambda_1$ and $\lambda_2$. Specifically, we selected the values for $\lambda_1$ and $\lambda_2$ from the set $\{0.1, 0.2, 0.3, 0.5, 1.0, 2.0\}$ based on their impact on the validation results, and we use $\lambda_1 = 1.0$ and $\lambda_2 = 0.2$.

As for STC (ShanghaiTech Campus) dataset [28], all training details and hyperparameters are the same, except for a learning rate that is set to be $(2e^{-4})$.

### B.2   Data Augmentation

- Small-sized rectangular shaped mask: $2 \le H \le 32$, $10 \le W \le 50$
- Medium-sized rectangular shaped mask: $16 \le H \le 64$, $64 \le W \le 256$
- Circular shaped mask: $10 \le radius \le 25$

To generate anomaly data, we first extracted patches based on the masks described above. These masks corresponded to small and medium-sized rectangular shapes, as well as circular shapes. Subsequently, the extracted patches underwent distortions to simulate anomalies, where we randomly perform color jittering between 0 and 0.1 (brightness, contrast, saturation, and hue). Additionally, we introduce random horizontal and vertical flips to further augment the dataset. These patches are then applied at the random locations of normal images from the same category. Figures S1 and S2 present categorical examples of synthetic anomaly images, with anomalies generated using rectangular patches in Figure S1 and circular patches in Figure S2.

We constructed a batch of size 8 by including an equal number of normal images and three types of abnormal images, with each type corresponding to a specific mask shape. This ensured a balanced representation of normal and abnormal samples within the batch, allowing for effective training of our model.

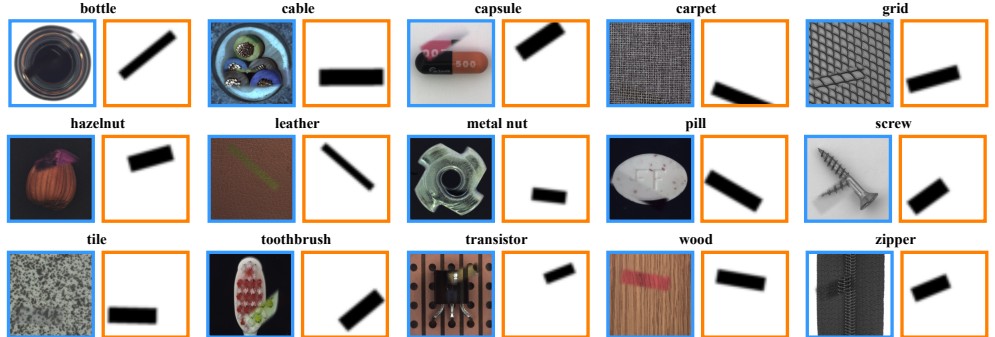

Figure S1: Synthetic anomaly images generated with rectangular shaped patches.

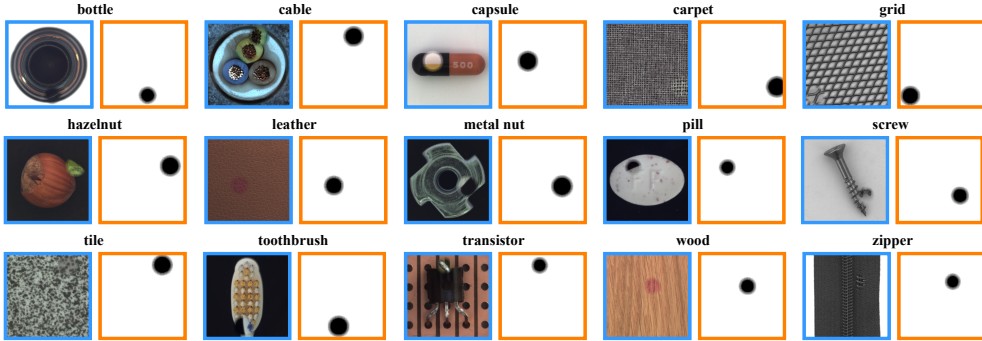

Figure S2: Synthetic anomaly images generated with circular shaped patches.

## C  Additional Experimental Results

### C.1  Qualitative Results

In Figure S3, we give additional qualitative results of anomaly score maps similar to Figure 4 in the main manuscript. Specifically, we compare against Model **(2a)** from Table 5, which is a variant of our model that lacks anomaly augmentation. Figure S3 illustrates that even without anomaly augmentation, Model **(2a)** is still able to localize anomaly regions, highlighting that the major effectiveness of our framework does not come from anomaly augmentations but from the capability to transform the distribution of features at each location of a given input image to different distributions. Regardless, the figure shows that our final model gives more accurate and clear anomaly score maps, owing to anomaly augmentation.

### C.2  Quantitative Results

In this section, we provide additional and detailed quantitative experimental results for Table 4, 5, and 6 in the main manuscript. In particular, we provide the performance of each category for Table 4, 5, and 6 in the corresponding tables: Table S2, S3, and S4, respectively. In Table S5 and S6, we provide additional ablation results, which are obtained by applying the proposed semantic-aware normalizing flow (NF) schemes to other NF-based methods: CFLOW-AD [20] and FastFlow [51]. The consistent performance improvement across CFLOW-AD and FastFlow demonstrates the flexibility and effectiveness of our proposed semantic-aware normalizing flow (NF) schemes.

### C.3  Anomaly Score Function

Unlike the loss function, we do not use $s(z_i)$ value in Equation 8 for computing anomaly score. However, $s(z_i)$ maybe helpful for defining better anomaly score function. To be sure that $s(z_i)$ is not helpful, we provide visualization examples Figure S4 and S5. Figure S4 represents the results during training, while Figure S5 demonstrates the results during test. As it can be observed in Figure S5,

during inference, there are instances where it tends to segment only localized regions or produce high scores in unrelated areas. This may be due to discrepancies between abnormal augmentation used during training and actual anomalies during test. Since, $s(z_i)$ is computed based on both negative log-likelihood (NLL) of normal and abnormal features, it is necessary for the model to project normal and abnormal features precisely. However, Figure S5 illustrates that the projection of abnormal features is not as accurate as as normal features, showing unsatisfactory results with $s(z_i)$ as a anomaly scoring function. Therefore, it does not exhibit suitability comparable to NLL for anomaly scoring. In the future, we have plans to utilize $s(z_i)$ for performance enhancement.

## C.4 Additional Analysis for Table 6

While estimating both mean and variance to give higher performance due to more flexibility gives a high expectation for the performances, we observe that giving too much flexibility and increasing training difficulty, thereby resulting in performance degradation as observed in Table 6 of the main paper and Figure S6. To better support the claim, we allow the model to estimate both mean and variance (i.e., Model (**3c**)) and plot the distribution of estimated mean by utilizing histogram in Figure S6. Graphs demonstrates that there are significant degree of overlap between normal (blue) and abnormal (orange) distributions, when considering variance as well.

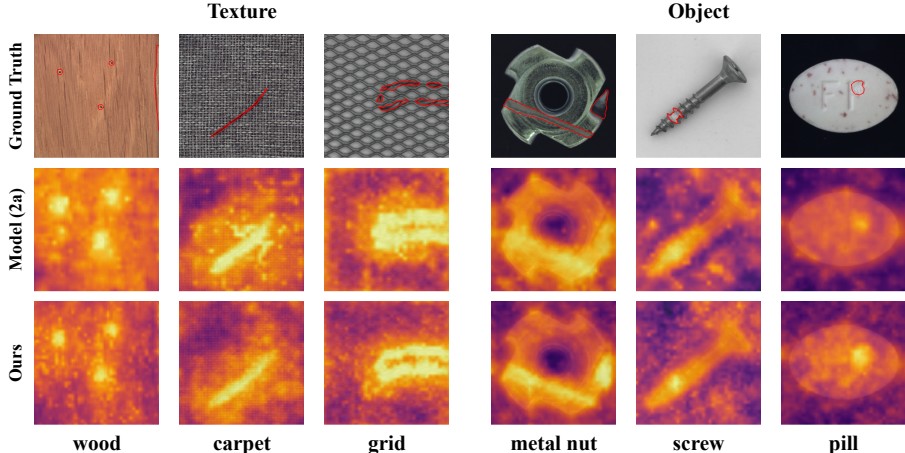

Figure S3: Visualization of anomaly score maps produced by our proposed framework and Model **(2a)** on texture and object categories of MVTec benchmark dataset. From top to bottom: GT mask, anomaly score map of proposed model and anomaly score map of Model **(2a)**.

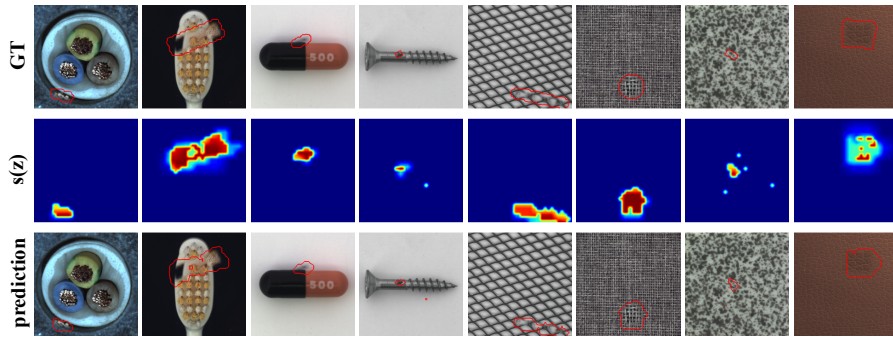

Figure S4: Ground truth, score map, and prediction for training data when using $s(z)$ as the anomaly score.

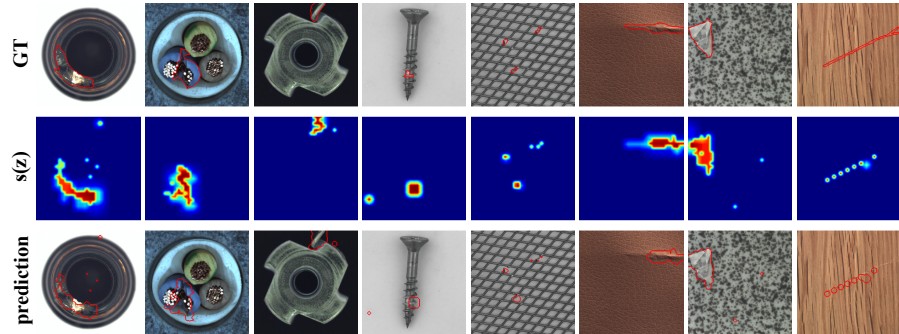

Figure S5: Ground truth, score map, and prediction for test data when using $s(z)$ as the anomaly score.

Table S1: Results by category in WRN-101 backbone.

| Category | bottle | cable | capsule | carpet | grid | hazelnut | leather | metal_nut | pill | screw | tile | toothbrush | transistor | wood | zipper | average |
|---|---|---|---|---|---|---|---|---|---|---|---|---|---|---|---|---|
| **Image-Wise** | 100 | 99.7 | 98.9 | 99.9 | 100 | 100 | 100 | 100 | 98.2 | 96.2 | 100 | 100 | 99.4 | 99.3 | 100 | 99.4 |
| **Pixel-Wise** | 99.1 | 98.8 | 98.9 | 99.4 | 99.3 | 99.0 | 99.8 | 98.7 | 99.1 | 99.2 | 99.1 | 99.2 | 95.1 | 97.9 | 99.6 | 98.8 |
| **Pro-Score** | 96.6 | 98.7 | 96.9 | 98.0 | 97.5 | 97.3 | 99.3 | 96.3 | 97.6 | 95.8 | 97.4 | 94.2 | 80.6 | 96.3 | 98.3 | 96.1 |

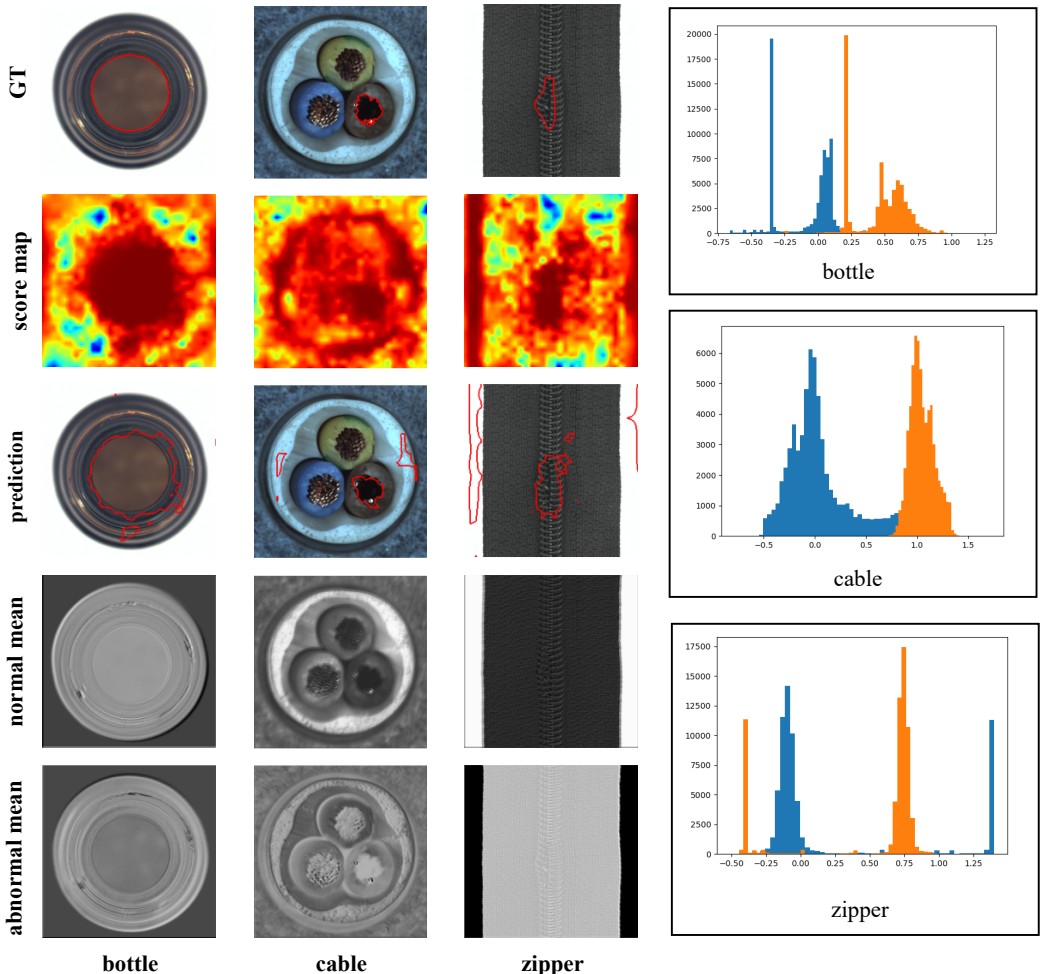

Figure S6: Results of Model **(3c)** (model that estimates both mean and variance), including ground truth, score map, prediction, normal mean estimated map, abnormal mean estimated map and histogram. For histogram plots, each displays estimated mean distribution of normal and abnormal features for each category: bottle, cable, and zipper. Orange part of histogram represents estimated abnormal means and blue part of histogram represents estimated normal means.

Table S2: Results by category in Table 4.

| Category | | bottle | cable | capsule | carpet | grid | hazelnut | leather | metal_nut | pill | screw | tile | toothbrush | transistor | wood | zipper | average |
|---|---|---|---|---|---|---|---|---|---|---|---|---|---|---|---|---|---|
| **Image-Wise** | (1b) | 100 | 97.7 | 97.4 | 99.8 | 99.6 | 100 | 100 | 99.3 | 96.0 | 90.5 | 100 | 98.3 | 98.2 | 97.4 | 99.8 | 98.2 |
| | (1c) | 100 | 96.7 | 98.2 | 99.7 | 99.5 | 100 | 100 | 99.5 | 96.6 | 89.0 | 100 | 97.8 | 98.8 | 98.6 | 99.8 | 98.3 |
| | (1d) | 100 | 98.2 | 97.5 | 99.8 | 99.7 | 100 | 100 | 99.5 | 96.2 | 92.8 | 100 | 95.3 | 99.1 | 98.9 | 99.9 | 98.5 |
| | **SANFlow** | 100 | 98.7 | 97.7 | 99.8 | 99.3 | 100 | 100 | 99.8 | 96.8 | 94.0 | 100 | 96.7 | 99.3 | 99.1 | 99.9 | 98.7 |
| **Pixel-Wise** | (1b) | 98.6 | 97.9 | 98.9 | 99.4 | 98.4 | 99.1 | 99.5 | 98.0 | 99.1 | 98.8 | 98.9 | 98.9 | 94.4 | 96.1 | 98.9 | 98.3 |
| | (1c) | 98.8 | 97.1 | 99.0 | 99.4 | 98.5 | 99.2 | 99.5 | 98.3 | 99.1 | 98.9 | 99.0 | 99.0 | 93.6 | 96.5 | 98.5 | 98.3 |
| | (1d) | 98.6 | 98.3 | 99.0 | 99.3 | 98.6 | 99.2 | 99.6 | 98.3 | 99.1 | 98.9 | 99.0 | 99.0 | 94.3 | 96.0 | 99.1 | 98.4 |
| | **SANFlow** | 98.6 | 98.5 | 99.1 | 99.3 | 98.5 | 99.2 | 99.6 | 98.5 | 99.2 | 99.0 | 98.9 | 98.9 | 94.4 | 96.4 | 98.9 | 98.5 |
| **Pro-Score** | (1b) | 92.8 | 95.8 | 92.9 | 96.7 | 94.6 | 96.3 | 98.2 | 87.8 | 94.7 | 94.4 | 93.9 | 87.5 | 88.1 | 90.9 | 95.5 | 93.3 |
| | (1c) | 94.1 | 94.8 | 93.9 | 96.5 | 94.2 | 96.6 | 98.5 | 91.7 | 95.3 | 94.9 | 94.4 | 89.2 | 87.8 | 91.2 | 94.6 | 93.8 |
| | (1d) | 93.1 | 96.5 | 93.9 | 96.5 | 94.6 | 96.8 | 98.5 | 91.9 | 94.9 | 94.7 | 94.7 | 89.3 | 88.4 | 91.4 | 96.2 | 94.1 |
| | **SANFlow** | 92.7 | 96.7 | 94.1 | 96.4 | 94.3 | 96.8 | 98.3 | 92.8 | 95.6 | 95.3 | 94.2 | 87.9 | 87.6 | 91.1 | 95.7 | 94.0 |

Table S3: Results by category in Table 5.

| Category | | bottle | cable | capsule | carpet | grid | hazelnut | leather | metal_nut | pill | screw | tile | toothbrush | transistor | wood | zipper | average |
|---|---|---|---|---|---|---|---|---|---|---|---|---|---|---|---|---|---|
| **Image-Wise** | (2a) | 100 | 97.5 | 97.2 | 99.4 | 99.2 | 100 | 100 | 98.9 | 97.4 | 93.2 | 99.9 | 98.0 | 96.6 | 99.0 | 99.1 | 98.4 |
| | SANFlow | 100 | 98.7 | 97.7 | 99.8 | 99.3 | 100 | 100 | 99.8 | 96.8 | 94.0 | 100 | 96.7 | 99.3 | 99.1 | 99.9 | 98.7 |
| **Pixel-Wise** | (2a) | 98.8 | 97.7 | 99.0 | 99.3 | 98.1 | 99.0 | 99.5 | 98.2 | 99.0 | 98.8 | 98.0 | 98.6 | 95.6 | 95.9 | 98.4 | 98.2 |
| | SANFlow | 98.6 | 98.5 | 99.1 | 99.3 | 98.5 | 99.2 | 99.6 | 98.5 | 99.2 | 99.0 | 98.9 | 98.9 | 94.4 | 96.4 | 98.9 | 98.5 |
| **Pro-Score** | (2a) | 94.1 | 92.6 | 93.5 | 96.5 | 93.0 | 96.1 | 98.8 | 92.0 | 94.3 | 94.3 | 92.3 | 87.6 | 85.3 | 93.1 | 92.9 | 93.1 |
| | SANFlow | 92.7 | 96.7 | 94.1 | 96.4 | 94.3 | 96.8 | 98.3 | 92.8 | 95.6 | 95.3 | 94.2 | 87.9 | 87.6 | 91.1 | 95.7 | 94.0 |

Table S4: Results by category in Table 6.

| Category | | bottle | cable | capsule | carpet | grid | hazelnut | leather | metal_nut | pill | screw | tile | toothbrush | transistor | wood | zipper | average |
|---|---|---|---|---|---|---|---|---|---|---|---|---|---|---|---|---|---|
| **Image-Wise** | (3a) | 100 | 98.2 | 97.4 | 99.3 | 100 | 100 | 100 | 99.3 | 95.3 | 87.1 | 100 | 95.8 | 94.4 | 99.5 | 98.0 | 97.6 |
| | (3b) | 100 | 94.9 | 98.6 | 99.4 | 86.9 | 100 | 100 | 99.2 | 96.0 | 89.4 | 100 | 97.5 | 95.1 | 99.7 | 98.0 | 97.0 |
| | (3c) | 100 | 98.0 | 96.6 | 99.7 | 99.9 | 100 | 100 | 99.8 | 95.4 | 89.3 | 99.9 | 95.6 | 98.6 | 99.0 | 99.9 | 98.1 |
| | SANFlow | 100 | 98.7 | 97.7 | 99.8 | 99.3 | 100 | 100 | 99.8 | 96.8 | 94.0 | 100 | 96.7 | 99.3 | 99.1 | 99.9 | 98.7 |
| **Pixel-Wise** | (3a) | 98.9 | 97.1 | 99.0 | 99.1 | 97.2 | 99.2 | 99.5 | 98.3 | 98.7 | 98.5 | 98.2 | 99.0 | 93.2 | 96.2 | 98.3 | 98.1 |
| | (3b) | 98.9 | 97.1 | 99.0 | 99.2 | 96.2 | 99.1 | 99.5 | 98.1 | 99.0 | 98.6 | 97.9 | 99.0 | 92.7 | 95.9 | 98.0 | 97.9 |
| | (3c) | 98.6 | 98.0 | 98.9 | 99.3 | 98.6 | 99.1 | 99.5 | 98.4 | 99.0 | 98.8 | 99.0 | 99.0 | 93.1 | 96.0 | 99.0 | 98.3 |
| | SANFlow | 98.6 | 98.5 | 99.1 | 99.3 | 98.5 | 99.2 | 99.6 | 98.5 | 99.2 | 99.0 | 98.9 | 98.9 | 94.4 | 96.4 | 98.9 | 98.5 |
| **Pro-Score** | (3a) | 94.8 | 92.6 | 94.3 | 95.4 | 88.4 | 96.8 | 98.5 | 92.8 | 95.4 | 93.9 | 91.9 | 90.8 | 80.9 | 92.1 | 91.2 | 92.7 |
| | (3b) | 95.2 | 90.7 | 93.2 | 95.4 | 91.8 | 96.0 | 97.8 | 90.6 | 95.1 | 92.6 | 93.1 | 87.7 | 86.2 | 90.8 | 93.9 | 92.7 |
| | (3c) | 93.3 | 96.0 | 93.5 | 96.5 | 94.6 | 97.0 | 97.9 | 90.9 | 94.5 | 95.1 | 92.9 | 90.6 | 84.6 | 91.2 | 96.2 | 93.7 |
| | SANFlow | 92.7 | 96.7 | 94.1 | 96.4 | 94.3 | 96.8 | 98.3 | 92.8 | 95.6 | 95.3 | 94.2 | 87.9 | 87.6 | 91.1 | 95.7 | 94.0 |

Table S5: Comparison for other normalizing flow models with statistics prediction (**SP** ✓) and without statistics prediction(**SP** ✗). Experiments are conducted with the same backbone network WRN-50 on MVTec and results are reported with respect to AUROC metric for two scenarios: image-wise anomaly detection and pixel-wise anomaly localization. For convenience, we represent **statistic prediction** as **SP** and **NF based methods** as **NF** in this table. Since there is no official code for FastFlow [51], we conduct experiments with a third party code (`https://github.com/openvinotoolkit/anomalib`).

| | NF | SP | bottle | cable | capsule | carpet | grid | hazelnut | leather | metal_nut | pill | screw | tile | toothbrush | transistor | wood | zipper | average |
|---|---|---|---|---|---|---|---|---|---|---|---|---|---|---|---|---|---|---|
| **Image-Wise** | CFLOW-AD | ✗ | 100 | 97.59 | 97.68 | 98.73 | 99.60 | 99.98 | 100 | 99.26 | 96.82 | 91.89 | 99.88 | 99.65 | 95.21 | 99.12 | 98.48 | 98.26 |
| | | ✓ | 100 | 97.63 | 97.17 | 99.44 | 99.21 | 100 | 100 | 98.92 | 97.38 | 93.20 | 99.93 | 98.03 | 96.62 | 99.04 | 99.11 | 98.37 |
| | FastFlow | ✗ | 100 | 96.2 | 96.3 | 99.4 | 100 | 99.4 | 99.9 | 99.5 | 94.2 | 83.9 | 100 | 83.6 | 97.9 | 99.2 | 95.1 | 96.3 |
| | | ✓ | 100 | 96.6 | 93.6 | 99.0 | 100 | 97.7 | 100 | 99.6 | 94.1 | 92.4 | 100 | 99.9 | 94.4 | 96.5 | 100 | 97.6 |
| **Pixel-Wise** | CFLOW-AD | ✗ | 98.76 | 97.64 | 98.98 | 99.23 | 96.89 | 98.82 | 99.61 | 98.56 | 98.95 | 98.10 | 97.71 | 98.56 | 93.28 | 94.49 | 98.41 | 97.87 |
| | | ✓ | 98.75 | 97.73 | 98.99 | 99.28 | 98.10 | 98.99 | 99.54 | 98.21 | 98.97 | 98.81 | 98.02 | 98.62 | 95.55 | 95.86 | 98.44 | 98.28 |
| | FastFlow | ✗ | 98.6 | 97.2 | 99.0 | 99.1 | 99.2 | 98.0 | 99.6 | 98.8 | 97.6 | 96.6 | 96.6 | 98.0 | 97.1 | 94.1 | 98.5 | 97.9 |
| | | ✓ | 98.7 | 97.4 | 99.0 | 99.3 | 99.1 | 98.4 | 99.6 | 98.5 | 99.4 | 98.0 | 98.9 | 98.5 | 95.1 | 97.6 | 98.9 | 98.4 |

Table S6: Detection and localization performances on STC dataset. **CFLOW-AD+SP** represents the results of CFLOW-AD with statistic prediction (SP) in Table S5.

| | (1b) | (1c) | (1d) | (2a) | (3a) | (3b) | (3c) | CFLOW-AD | CFLOW-AD+SP | SANFlow |
|---|---|---|---|---|---|---|---|---|---|---|
| **Image-Wise** | 73.5 | 73.1 | 74.2 | 73.6 | 75.9 | 75.1 | 76.1 | 72.63 | 73.6 | **76.1** |
| **Pixel-Wise** | 91.5 | 91.5 | 94.6 | 94.5 | 94.6 | 94.1 | 93.7 | 94.48 | 94.5 | **94.8** |

