# OpenReview forum: "SANFlow: Semantic-Aware Normalizing Flow for Anomaly Detection"
_NeurIPS.cc/2023/Conference — NeurIPS 2023 poster_

### Official Review · Reviewer_cBPv · 2023-07-01

**Soundness:** 3 good
**Presentation:** 3 good
**Contribution:** 1 poor
**Rating:** 5
**Confidence:** 3

**Summary:**

The manuscript proposes a new method for dense anomaly detection in industrial applications. The proposed method first extracts dense features with a frozen pretrained model. Then, a normalizing flow learns the distribution of extracted features at every spatial location. While the flow parameters are shared across the spatial locations, the parameters of flow prior are independently computed by an additional network h conditioned on the input image. In particular, the network h outputs variances of flow priors while means remain fixed. During the training, the input samples are augmented with synthetic negatives. The optimization objective is formed of the likelihood maximization for inlier features,  regularization of network h outputs, and discrimination between inliers and synthetic negatives. During the inference, the method detects anomalies according to the aggregated likelihood of features across different resolutions. The developed method achieves competitive results on two considered benchmarks.

**Strengths:**

S1. The manuscript is well-written and easy to follow.

S2. Modelling different normalizing flow prior at every spatial location appears to be novel.

S3. The proposed method achieves performance gains over related works.

**Weaknesses:**

W1. The difference between CFlow-AD [19] and the proposed SANFlow i) is modelling independent flow priors for each spatial location and ii) mapping anomalous and non-anomalous features to flow priors with different means. These contributions may be rather incremental.

W2. Intuition on why estimating both mean and variance is inferior to estimating variance only (L197) should be comprehensively elaborated beyond L323-328. Especially since predicting parameters of flow prior is the main contribution of the manuscript.

W3. Missing related works which augment datasets with synthetic negatives [a,b,c,d].

[a] Shu Kong, Deva Ramanan: OpenGAN: Open-Set Recognition via Open Data Generation. ICCV 2021.

[b] Matej Grcic, Petra Bevandic, Sinisa Segvic:
Dense Open-set Recognition with Synthetic Outliers Generated by Real NVP.  VISAPP 2021.

[c] Victor Besnier, Andrei Bursuc, David Picard, Alexandre Briot: Triggering Failures: Out-Of-Distribution detection by learning from local adversarial attacks in Semantic Segmentation. ICCV 2021.

[d] Kimin Lee, Honglak Lee, Kibok Lee, Jinwoo Shin:
Training Confidence-calibrated Classifiers for Detecting Out-of-Distribution Samples. ICLR 2018

**Questions:**

Adding a colorbar to Fig. 4 might improve clarity.


**Limitations:**

The manuscript adequately addressed the limitations of the method.

---

> ### Author Rebuttal · Authors · 2023-08-09
>
> Thank you for providing a thoughtful review.
> For enhancing our paper, we have diligently reviewed the weaknesses and questions raised regarding our paper and have prepared additional experiments and answers.
>
> **W1: Novelty**
> * We cautiously wish to assert the novelty of our approach. Contrary to previous Normalizing Flow (NF)-based frameworks, which use the same base distribution for all features and thus suffer from learning difficulty, our framework addresses this non-trivial issue by making full use of semantic features to dynamically map each feature to their respective base distributions, thereby increasing anomaly detection performance (as also highlighted by Reviewer dSr9).
> * Training NF to map each feature to its own respective base distributions is non-trivial. We introduce a statistical prediction network and our proposed loss functions (**Eq. (4) and (8)**) to guide the training of different semantic features according to their respective distributions (as also highlighted by Reviewer 7gCM).
> * Furthermore, we would kindly refer the reviewer to **L49-55** in the main paper for novelty of our method. In the context of vision anomaly detection, we firmly believe that estimating spatially and semantically varying base distributions based on the semantic information of features, enhancing discriminability and thereby anomaly detection performance, constitutes meaningful contributions.
>
> **W2: Why estimating both mean and variance is inferior to estimating only variance**
> * We agree that we need more explanation or intuition of **L323-328**.
> While one might expect estimating both mean and variance to give higher performance due to more flexibility, we observe that this gives too much flexibility and increases training difficulty, as observed in Table 6 of the main paper and **Figure C** in PDF attached above.
> To better support our claim, we allow the model to estimate both mean and variance (i.e., **Model (3c)**) and plot the distribution of estimated mean in **Figure D** in PDF attached above.
> The figure demonstrates that there are significant degree of overlap between normal (blue) and abnormal (orange) distributions, when considering variance as well.
> We will include the discussions in the revised paper.
>
> **W3: Related works on data augmentation with synthetic negatives**
> * Thanks to your kind suggestion, we believe we can incorporate missing references. We will make sure to include the papers [a, b, c, d].
>
> **Q1: Adding a color bar**
> * Lastly, we appreciate the advice mentioned in the question. We will also add a color bar to Figure 4 to enhance clarity.

---

> > ### Comment · Reviewer_cBPv · 2023-08-16
> > **Post rebuttal**
> >
> > The authors resolved my concerns hence I increase the score to BA.

---

> > > ### Author Response · Authors · 2023-08-18
> > >
> > > We appreciate the reviewer's time and constructive comments and discussions, which have definitely improved our paper. We are thankful for acknowledging our rebuttal as well. We will include discussions in the final version.

---

### Official Review · Reviewer_dSr9 · 2023-07-05

**Soundness:** 3 good
**Presentation:** 3 good
**Contribution:** 2 fair
**Rating:** 5
**Confidence:** 5

**Summary:**

This paper presents a novel NF-based framework for anomaly detection (AD) and anomaly localization (AL) task. The proposed Semantic-Aware Normalizing Flow (SANFlow) framework map the distributions of normal data to different distributions at each location in the given image instead of mapping a distribution of whole image and makes distributions of abnormal data distant from normal data. The proposed method achieves the state-of-the-art in both AD and AL.

**Strengths:**

[1. The proposed SANFlow model addresses the problem of learning difficulty in anomaly detection, which is conventional in NF-based methods. This innovation makes full use of semantic information and enhances the accuracy of anomaly detection.
2. This paper provides a detailed description of the proposed SANFlow model. The methodology is well-developed and supported by theoretical analysis.
3. The experimental results on the Mvtec-AD demonstrate the superiority of SANFlow.

**Weaknesses:**

1. This paper lacks the implementation details of the multi-scale FN, making it difficult to reproduce the study. And no code is provided to reproduce the proposed method.
2. This paper needs ablation studies about the margin between normal and abnormal means.
3. More experiments should be conducted. The paper lacks experiments on other standard AD datasets like CIFAR10.

**Questions:**

1.	Can the authors provide more details on the implementation of the SANFlow model, including the multi-scale FN?
2.	Can you explain why no code implementation is provided, since open-source is necessary for the community?
3.	Have the authors considered comparing the performance of the proposed model on other datasets, such as CIFAR10?

**Limitations:**

Yes.

---

> ### Author Rebuttal · Authors · 2023-08-09
>
> We sincerely appreciate Reviewer dSr9 (R3) acknowledgment of the well-developed methodology of our paper and the recognition of the experimental results showcasing the superiority of the proposed method.
> In order to facilitate a deeper understanding, we have also prepared responses addressing the points the Reviewer mentioned weaknesses of our paper.
>
> **W1: Multi-scale NF details and code**
> * To implement multi-scale NF, we utilized the FrEIA library.
> * For reproducibility, along with implementation details provided in **L263-269** in our main paper and **Section B** in the supplementary document, we will release our code upon acceptance.
>
> **W2: Mean margin ablation**
> * To enhance the persuasiveness of the mean margin, we conducted additional experiments on the MVTec-AD dataset, comparing performance at margin values of 0.5, 1.0, and 1.5 in  WRN-50 backbone.
> * As the margin of 1.0 yielded the highest results, we could confirm a mean margin of 1.0 is a persuasive choice.
> * We believe this is due to trade-off between discriminability and representation. The anomaly and normal features have subtle differences. If the margin difference is too small, the model will have difficulty discriminating normal and abnormal features. If the margin difference is too large, the model is forced to impose large differences on normal and abnormal features that have very small and subtle differences, increasing training difficulty.
> | Margin     | 0.5  | 1.0  | 1.5  |
> |------------|------|------|------|
> | Image wise | 98.6 | 98.7 | 98.1 |
> | Pixel wise | 98.4 | 98.5 | 98.3 |
>
> **W3: Experiments on other datasets**
> * While our approach excels at detecting anomalies within images, its effectiveness in tasks involving image-wise semantic outlier detection, like CutPaste and CFLOW-AD, is limited. As a result, specific results for these tasks were not included in the paper. Nonetheless, we conducted an additional experiment with CIFAR-10, where our proposed method achieved an AUROC of 80.8, outperforming CutPaste (69.4) and CFLOW-AD (79.32).
> * In the paper, there are results on MVTec-AD and STC dataset and in response to **7gCM W2** we provide results on the VisA dataset.

---

### Official Review · Reviewer_VpXm · 2023-07-05

**Soundness:** 3 good
**Presentation:** 3 good
**Contribution:** 2 fair
**Rating:** 6
**Confidence:** 4

**Summary:**

This paper propose a NF-based framework to model the normal feature distribution as Gaussian distributions with zero mean but different variances, and the abnormal features as distinct Gaussian distribution. The proposed model demonstrates relatively good performances in both anomaly detection and localization.

**Strengths:**

The writing of this paper is clear and concise. It's easy to follow and understand each component of the proposed model. The motivation to model various distributions according to different regoins are reasonable.

**Weaknesses:**

1. The novalty of this paper seems to be increamental. It's based on / combanition of the reference papers [34] [22] [19] [48].
2. The incorporation of semantic information seems to be naive. All the pixels have the same base distributions following the inverse
Gamma distribution as IG(·|α, β).  I wonder if the semantic information can be incorparated into the corresponding base distributions (The same regoin share the same base distribution).
3.  I don't understand why you fix µi to be 0 for normal regions and 1 for abnormal regions. As the normal pixels have various characteristics, maybe corresponding to various µi, at least fot different regoins.
4.  s(zi) maybe helpful for computing anomaly score. I'd like to see the correlations between s(zi)  and  GT location, which can reflect the gap between simulated training  anomalies and real testing anomalies.

**Questions:**

Is the performance gain of 0.1% brought by anomaly augmentation? It seems the simulated anomalies are meaningless.

**Limitations:**

Yes

---

> ### Author Rebuttal · Authors · 2023-08-08
>
> We are glad to hear that you found the paper to be clear and concise. We have diligently examined your comments and concerns as a reviewer, and have prepared responses addressing the raised concerns.
>
> **W1: Novelty**
> * We cautiously wish to assert the novelty of our approach. Contrary to previous Normalizing Flow (NF)-based frameworks, which use the same base distribution for all features and thus suffer from learning difficulty, our framework addresses this non-trivial issue by making full use of semantic features to dynamically map each feature to their respective base distributions, thereby increasing anomaly detection performance (as also highlighted by Reviewer dSr9).
> * Training NF to map each feature to its own respective base distributions is non-trivial. We introduce a statistical prediction network and our proposed loss functions (**Eq. (4) and (8)**) to guide the training of different semantic features according to their respective distributions (as also highlighted by Reviewer 7gCM).
> * Furthermore, we would kindly refer the reviewer to **L49-55** in the main paper for novelty of our method. In the context of vision anomaly detection, we firmly believe that estimating spatially and semantically varying base distributions based on the semantic information of features, enhancing discriminability and thereby anomaly detection performance, constitutes meaningful contributions.
>
> **W2: Naive incorporation of semantic information, all pixels share the same base distribution?**
> * Please allow us to provide a clarification of the proposed model: Each feature (or pixel) is mapped to its own respective base distribution with estimated variances based on the semantic characteristics of individual features, as illustrated in **Figure 1** of the main paper.
> In essence, *all pixels **do not** follow the same base distribution* but follow its own respective base distributions, each with its own statistics (i.e., variance in our case) estimated by statistics prediction network, conditioned on semantic information.
> * The inverse Gamma distribution, IG($\cdot$|$\alpha$,$\beta$), is used to guide the generation of $\alpha_i$,$\beta_i$ by the statistics prediction network for regularization and stable training, as mentioned in **L209-213** of our main paper.
> But, $\alpha_i$,$\beta_i$ will not be equal to $\alpha$,$\beta$, as the overall framework, including the statistics prediction network, is trained to optimize our overall loss function (**Eq. (9)**), which also includes NLL. Since features are mapped to base distributions according to variances following IG($\cdot$|$\alpha_i$,$\beta_i$) and $\alpha_i$,$\beta_i$ vary depending on semantic features, features are not mapped to the same base distribution.
> * In addition, the large performance gap between only using same base distribution Model **(1a)** and using spatial varying distributions (**SANFlow**) shown in Table 4 demonstrates the effectiveness of our framework.
>
> **W3: Why the means were fixed**
> * While one might expect estimating both mean and variance to give higher performance due to more flexibility, we observe that this gives too much flexibility and increases training difficulty, thereby resulting in performance degradation, as observed in Table 6 of the main paper and **Figure C** in PDF attached above.
> To better support our claim, we allow the model to estimate both mean and variance (i.e., **Model (3c)**) and plot the distribution of estimated mean in **Figure D** in PDF attached above.
> The figure demonstrates that there are significant degree of overlap between normal (blue) and abnormal (orange) distributions, when considering variance as well.
>
> **W4: Using $s(z)$ for anomaly score calculation**
> * We have provided visualization examples in the attached PDF. **Figure A** represents represent the results during training, while **Figure B** demonstrates the results during test.
> * As it can be observed in **Figure B**, **during inference, there are instances where it tends to segment only localized regions or produce high scores in unrelated areas**.
> This may be due to discrepancies between abnormal augmentation used during training and actual anomalies during test.
> Since, $s(z)$ is computed based on both NLL of normal and abnormal features, it is necessary for the model to project normal and abnormal features precisely. However, **Figure B** illustrates that the projection of abnormal features is not as accurate as as normal features, showing unsatisfactory results with $s(z)$ as a anomaly scoring function.
> Therefore, it does not exhibit suitability comparable to NLL for anomaly scoring. Furthermore, we explored various attempts, such as multiplying or adding $s(z)$ to the existing anomaly score, yet the results favored using NLL alone. In the future, we have plans to utilize $s(z)$ for performance enhancement.
>
> **Q1: The performance improvement due to anomaly augmentation**
> * Before answering the question, we would like to emphasize that our major contribution involves finding appropriate distribution of different semantic features and also mapping anomaly feature to completely different base distribution.
> * To train model to map anomaly feature to a distant distribution, an augmentation process is necessary.
> * For the answer, we agree that 0.1 seems a small improvement at MVTec-AD dataset.
> As the performance of MVTec-AD is saturated, we considered the STC dataset results for models (Model **(2a)**) and (Model **(1d)**) to show assistance effect of anomaly augmentation.
> For STC, we observed a performance improvement from 73.6 (Model **(2a)**) to 74.2 (Model **(1d)**).
> * Anomaly augmentation allows us to use BCE loss for support mapping anomaly features to the distribution distant from the normal base distribution.
> **Table 5** and **Fig. S3** in supplementary include find qualitative and quantitative difference with and without BCE loss, highlighting the importance of using anomaly augmentation.

---

> > ### Comment · Reviewer_VpXm · 2023-08-16
> >
> > Thank the authors for their effort in response. Most of my questions have been addressed. However, I think the authors might misunderstand the `base distribution' mentioned in W2, where the IG( , | alpha,beta) refers to the second term in the equation 4, rather than the first term.

---

> > > ### Author Response · Authors · 2023-08-18
> > > **Our response to the reviewer VpXm:**
> > >
> > > **[Acknowledging clarification]**
> > > * Thank you for the response and clarification. We now understand that the base distribution mentioned by the reviewer VpXm refers to the base distribution for variances. To avoid confusion with base distribution for NF, **we will refer to the mentioned base distribution as the *prior distribution* for variance**, denoted as IG($\cdot$|$\alpha$,$\beta$) in Eq(4) in the paper. Please correct us if we misunderstood.
> > >
> > >
> > > **[Semantic-dependent prior distributions]**
> > > * We agree with the reviewer that if we can incorporate semantic information into the prior distribution for variances, we can better facilitate the training of NF by guiding the statistics prediction network $h$ to predict variances close to the pre-determined prior distribution while minimizing negative log-likelihood (NLL). **However, the challenge lies in finding such prior distribution that is specific to each semantic information or spatial region.**  Even at the same pixel location, we can have different semantic information or even for the same category due to misalignment.
> > >
> > >
> > > **[Justification for using the fixed prior distribution]**
> > > * Prior distribution is usually pre-determined with prior knowledge as to how distribution should be. And, for each image and spatial region, we have to rely on semantic information to determine how the distribution should be. But, **we lack such prior knowledge regarding semantic-dependent prior distribution.** Therefore, instead of heuristically finding the semantic-dependent prior distribution, **we defined a single prior distribution for all pixels**, where we set $\alpha = \frac{p^2}{2} - 1$, where $p^2$ represents the area of receptive field for $h$ since $\alpha$ represents the extent to which information about the predicted variance is reflected. Meanwhile, $\beta$ is set to be $\frac{p^2\xi}{2}$, where $\xi$ is the mode value of IG (since the mode of IG can be found by $\frac{\beta}{\alpha+1}$). $\xi$ is a hyperparameter whose value we empirically set to be 0.1. If $\xi$ is too high, this causes too much overlap between base distributions for normal and abnormal features, and if $\xi$ is too low, this decreases the diversity of base distributions.
> > >
> > >
> > > **[Attempts on semantic-dependent prior distribution]**
> > > * Regardless, since we agree that we can improve the training of NF if we can successfully incorporate semantic information into the prior distribution, we tried to derive semantic-dependent prior distribution using statistics of semantic features, such as their mean and variance. However, we could not obtain improvements.
> > >
> > >
> > > **[Still, our framework is semantic-aware NF]**
> > > * Nonetheless, as discussed before, **our semantic-aware NF framework makes use of semantic features to dynamically map each feature to their respective base distributions**, greatly improving the performance of anomaly detection, while previous NF-based works employed the same base distributions for all features. We were able to achieve this by jointly training statistics prediction network $h$ and NF model to minimize NLL loss, along with two introduced loss terms **Eq. (4)** and **Eq. (8)**. NLL loss will encourage NF model and our statistics prediction network to find respective base distributions for each feature while **Eq. (4)** regularizes and guides the prediction of variances and **Eq. (8)** guides our framework to distinguish normal and abnormal features.
> > >
> > >
> > > * We would like to further express our gratitude to the reviewer for the constructive comments and insightful discussions, which have definitely improved our work. We will add the discussions in the final version.

---

> > > > ### Comment · Reviewer_VpXm · 2023-08-20
> > > >
> > > > Thanks for your response and clarification. I'd like to increase my score to weak accept.

---

> > > > > ### Author Response · Authors · 2023-08-20
> > > > > **Our response to the reviewer VpXm:**
> > > > >
> > > > > We value the reviewer VpXm's time and interest in reviewing our paper. We also appreciate the reviewer VpXm for improving our paper's score. The provided feedback significantly helped solidify the paper and will be included in the future final version.

---

### Official Review · Reviewer_7gCM · 2023-07-06

**Soundness:** 3 good
**Presentation:** 3 good
**Contribution:** 2 fair
**Rating:** 5
**Confidence:** 5

**Summary:**

This paper presents an enhanced flow-based anomaly detection method by utilizing density estimation on normalizing flow (NF). The proposed approach involves predicting the inverse Gamma distribution using a statistics prediction network, which facilitates dynamic training of normalized flow distribution. Additionally, the training process benefits from the introduction of anomaly feature distribution and incorporates data augmentation techniques to enhance performance. By learning to map diverse normal features and abnormal features to distinct base distributions, the proposed NF framework enhances its capability for density estimation. The experimental results show that the improved density modeling achieved by the proposed framework, resulting in enhanced anomaly detection performance.

**Strengths:**

+ In contrast to previous flow-based anomaly detection methods that transform the distribution of all features into a single distribution, this work introduces a statistical prediction network to guide the training of different semantic features according to their respective distributions.
+ By incorporating the data augmentation, both normal and anomaly distribution are predicted and employed to derive to the anomaly score. The qualitive results demonstrate that the localization performance of the proposed method is improved by data augmentation.


**Weaknesses:**

- The paper does not provide the detection performance of the PRO metric. The flow-based method tends to produce oversized defect regions, particularly for small-sized defects. Therefore, it is crucial to evaluate the location performance using the PRO metric.
- For image based anomaly detection, several recent AD methods achieves promising the image level and pixel AUROC on MVTEC AD, which make it difficult to identify the advantage while comparison. Therefore, it is necessary to compare the performance on other datasets, e.g. VisA. In VisA, images exhibit complex structures, objects placed in sporadic locations, and different objects, which could differentiate the detection performance of different methods.
- The paper utilizes a self-defined data augmentation method to synthesize anomalies. However, several existing data augmentation methods, such as Draem, nsa, and cutpaste, have been proposed for anomaly detection. It is important to consider whether the choice of data augmentation method impacts the detection performance.


**Questions:**

Please see the weaknesses section. The comprehensive results on image-based anomaly detection are expected.

**Limitations:**

Authors have addressed the limitations of the method.

---

> ### Author Rebuttal · Authors · 2023-08-09
>
> Thank you very much for thoroughly reviewing our paper.
> We appreciate your feedback. To address the concerns you raised, we are providing several experimental results and our perspectives.
>
> **W1: PRO metric**
> * In response to the question, we have taken the initiative to provide the PRO scores with WRN50 backbone in Supplementary **Table** **S1**.
> It is noteworthy that the method utilizing Wide-ResNet50 as a backbone exhibits a commendable PRO score of 94.0 and also with an additional experiment with Wide-ResNet101 as a backbone we kindly provide a PRO-score of 96.0.
> * This performance not only stands out when compared to alternative methodologies that could serve as reference points, but also underscores its inherent merit by showcasing its immunity to the challenges associated with flow-based models.
>
> **W2: VisA dataset**
> * Another weakness Reviewer 7gCM (R1) mentioned was about the difficulty in identifying the advantage with the MVTec-AD dataset.
> Furthermore, R1 thankfully advised that a performance comparison in the VisA dataset was necessary.
> To instill confidence in our approach, we conducted additional experiments on the VisA.
> We actually conducted experiment for not only ours, but also DRAEM, CFLOW-AD, and PaDiM.
> In addition, since our method is a flow-based network, we added results of FastFlow in paperwithcode for comparison.
>
> | VisA      | Image AUROC | Pixel PRO |
> |-----------|-------------|-------------|
> | Ours (NF-based)      | **93.4**        | **89.4**        |
> | CFLOW-AD (NF-based) | 91.5        | 59.8        |
> | FastFlow (NF-based) | 82.2        | 59.8        |
> | DRAEM     | 88.7        | 73.1        |
> | PaDiM     | 89.1        | 85.9        |
>
> * Our proposed method, with a WRN50 backbone, achieved an image, pixel-wise AUROC of 93.4, 98.6, and a PRO score of 89.4.
> This shows that our approach demonstrates competitive performance across all methods, including other flow-based approaches, demonstrating the effectiveness of our framework in greatly improving the density estimation capability of Normalizing Flow for anomaly detection.
>
> **W3: Data augmentation ablation**
> * We employ the CutPaste technique with blurred boundaries to create more realistic abnormal images.
> * However, in response to the Reviewer's concerns, we undertook an additional performance comparison by incorporating the augmentation approach of DRAEM.
> When applying the DRAEM augmentation method for image-wise anomaly detection, we achieved an AUROC score of 97.1 while for pixel-wise anomaly detection, the AUROC score was 96.4.
> These results indeed suggest that the augmentation method from DRAEM provides very complex abnormal regions and does not appear to yield positive contributions within the framework of our methodology, as indicated by the observed performance disparities.
> * While we need appropriate anomaly augmentation to properly map abnormal features to different base distributions, we show that our model can still work and perform better than a baseline (**Model (1a)** in Table 4) when we do not use anomaly augmentation at all (**Model (2a)**) in Table 5.
> While baseline **Model (1a)** gives 72.6 image-wise AUROC for STC dataset and 98.3 for MVTec dataset, our framework without anomaly augmentation **Model (2a)** gives 73.6 image-wise AUROC for STC dataset and 98.4 for MVTec dataset.
> The performance improvement without anomaly augmentation suggests the effectiveness and importance of dynamically mapping features to its own respective base distributions based on semantic information.
> Regardless, learning to map abnormal features to different base distributions from normal features without anomaly augmentations or while being robust to choice of anomaly augmentations is important yet challenging research topic, which we leave as our future work.

---

### Author Rebuttal · Authors · 2023-08-09

To the reviewers.

First of all, we are grateful for the reviewers' efforts and we have prepared responses to comments and tables for additional experiments. For better explanations, we include visualizations in PDF attached below. We provide brief explanations for each figure:

There are a total of four figures labeled as A, B, C, and D.
* **Figure A**:
    * First three rows
    * Depicting ground truth, score map, and prediction for ***training data*** when using s(z) as the anomaly score.
    * Referred in response to **Reviewer** **VpXm** **W4**
* **Figure B**:
    * Next three rows
    * Depicting ground truth, score map, and prediction for ***test data*** when using s(z) as the anomaly score.
    * Referred in response to **Reviewer** **VpXm** **W4**
* **Figure C**:
    * Last five rows of images on the left, where each column corresponds to different examples.
    * Results of **Model (3c)** (model that estimates both mean and variance) , including ground truth, score map, prediction, normal mean estimated map, and abnormal mean estimated map
    * Referred in response to **Reviewer** **VpXm** **W3** and **Reviewer** **cBPv** **W2**
* **Figure D**:
    * Three figures in the bottom right corner
    * Each histogram displays estimated mean distribution of normal and abnormal features for each category: bottle, cable, and zipper.
    * Orange part of histogram represents estimated abnormal means and blue part of histogram represents estimated normal means.
    * Referred in response to **Reviewer** **VpXm** **W3** and **Reviewer** **cBPv** **W2**

---

### Decision · Program_Chairs · 2023-09-21

**Decision:**

Accept (poster)

**Comment:**

This paper got slightly positive but reserved reviews: one "weak accept" and three "borderline accept".

The authors addressed the reviewers' concerns by elaborating on the novelty and providing additional results, including evaluating with PRO metric, testing on the VisA dataset, adopting the augmentation approach of DRAEM, and experimenting with different margin values. The authors had further discussions with **Reviewer VpXm** on the issue of incorporating semantic information. After the rebuttal, **Reviewer VpXm** and **Reviewer cBPv** increased their ratings.

In more detail, the authors' rebuttal has addressed the following concerns raised in the reviews:
1. Novelty: The authors emphasize that previous Normalizing Flow (NF)-based methods on anomaly detection use the same base distribution for all features while their method maps semantic features to their respective base distributions and thus achieves better performance. The authors introduce a statistical prediction network with tailored loss functions to guide the training. The authors' discussions with **Reviewer VpXm** further clarify that it is challenging to find semantic-dependent prior distributions for variances. **Reviewer VpXm** has increased the rating to weak accept after discussions.
2. Experiments on the detection performance with the PRO metric: The supplementary material of the initial submission has presented PRO scores using Wide-ResNet50 as a backbone. The authors further present in the rebuttal an additional result with Wide-ResNet101 as the backbone, achieving a PRO-score of 96.0.
3.  VisA dataset: The authors provide additional experiments on the VisA dataset in the rebuttal. With a WRN50 backbone, the proposed method achieves AUROC scores of 93.4 (image-level), 98.6 (pixel-level), and a PRO score of 89.4, which are better than those of CFLOW-AD, FastFlow,
DRAEM, and PaDiM.
4. Augmentation: The authors present an additional comparison to show that adopting the augmentation approach of DRAEM does not improve the proposed method: with DREAM's augmentation, it achieves an AUROC score of 97.1 for image-wise anomaly detection and 96.4 for pixel-wise anomaly detection.
5. Ablation: The rebuttal also provides additional ablation studies about the margin (0.5, 1.0, and 1.5) between normal and abnormal means.

After reading the paper, reviews, author feedback, and discussions, the AC does not notice significant flaws that lead to overruling the reviewers' assessment. Furthermore, the AC acknowledges that the quality and contribution of this paper will be significantly improved if the authors elaborately revise the paper with respect to the reviews and discussions.
The AC thus recommends accepting the paper and trusts that the authors, in good faith, will improve the paper to address the reviewers' comments and incorporate the additional results when preparing the final version.